# Structural basis for Parkinson's disease-linked LRRK2's binding to microtubules

**David M. Snead**[1,2,3,8], **Mariusz Matyszewski** [1,2,4,8], **Andrea M. Dickey** [1,2,8], **Yu Xuan Lin**[1,2], **Andres E. Leschziner** [1,2,5,9]✉ **& Samara L. Reck-Peterson** [1,2,6,7,9]✉

Leucine-rich repeat kinase 2 (*LRRK2*) is one of the most commonly mutated genes in familial Parkinson's disease (PD). Under some circumstances, LRRK2 co-localizes with microtubules in cells, an association enhanced by PD mutations. We report a cryo-EM structure of the catalytic half of LRRK2, containing its kinase, in a closed conformation, and GTPase domains, bound to microtubules. We also report a structure of the catalytic half of LRRK1, which is closely related to LRRK2 but is not linked to PD. Although LRRK1's structure is similar to that of LRRK2, we find that LRRK1 does not interact with microtubules. Guided by these structures, we identify amino acids in LRRK2's GTPase that mediate microtubule binding; mutating them disrupts microtubule binding in vitro and in cells, without affecting LRRK2's kinase activity. Our results have implications for the design of therapeutic LRRK2 kinase inhibitors.

PD is the second most common neurodegenerative disease, affecting more than ten million people worldwide. Autosomal dominant missense mutations in *LRRK2* are a major cause of familial PD[1–4], and mutations in *LRRK2* are also linked to sporadic cases of PD[5,6]. All PD-linked mutations in LRRK2 increase its kinase activity[7–10], and increased LRRK2 kinase activity in the context of a wild-type (WT) protein is also associated with sporadic PD cases[11]. LRRK2-specific kinase inhibitors have been developed to treat PD and are in clinical trials (NCT04056689 and NCT03710707).

Although it remains unclear how it drives PD, LRRK2 has been functionally linked to membrane trafficking[12–14]. Mutant LRRK2 causes defects in endo/lysosomal, autophagosomal, and mitochondrial trafficking[15–19], and LRRK2 regulates lysosomal morphology[20–23]. Although the bulk of LRRK2 is found in the cytosol, it can associate with membranes under some conditions[20,21,24–27]. A subset of Rab GTPases, which

are master regulators of membrane trafficking[28], are phosphorylated by LRRK2, and PD-linked LRRK2 mutations increase Rab phosphorylation in cells[9,29]. Phosphorylation of Rabs by LRRK2 is linked to alterations in ciliogenesis[25,26,28] and defects in endolysosomal trafficking[16,20,22,23,30–32]. LRRK2 also co-localizes with microtubules in cells and in vitro[33–36]. Cellular localization of LRRK2 with microtubules is seen with elevated expression levels and is enhanced by type-1 LRRK2-specific kinase inhibitors[33,35,37,38]. In vitro, the catalytic half of LRRK2 alone can bind to microtubules[35]. In addition, many PD-linked mutations (p.R1441C, p.R1441G, p.Y1699C, and p.I2020T) increase microtubule association in cells in conjunction with elevated expression of the mutant protein[33,37]. It is not understood how LRRK2 perturbs cellular trafficking or how the cellular localization of LRRK2—cytosolic, membrane-associated, and/or microtubule-bound—contributes to its function or PD pathology. Developing tools that control the localization of LRRK2 in cells is

[1]Department of Cellular and Molecular Medicine, University of California, San Diego, La Jolla, CA, USA. [2]Aligning Science Across Parkinson's (ASAP) Collaborative Research Network, Chevy Chase, Maryland, MD, USA. [3]Department of Biochemistry and Molecular Biology, Johns Hopkins University Bloomberg School of Public Health, Baltimore, MD, USA. [4]School of Biological Sciences, University of California, San Diego, La Jolla, CA, USA. [5]Department of Molecular Biology, School of Biological Sciences, University of California, San Diego, La Jolla, CA, USA. [6]Department of Cell and Developmental Biology, School of Biological Sciences, University of California, San Diego, La Jolla, CA, USA. [7]Howard Hughes Medical Institute, Chevy Chase, Maryland, MD, USA. [8]These authors contributed equally: David M. Snead, Mariusz Matyszewski, Andrea M. Dickey. [9]These authors jointly supervised this work: Andres E. Leschziner, Samara L. Reck-Peterson. ✉e-mail: aleschziner@health.ucsd.edu; sreckpeterson@health.ucsd.edu

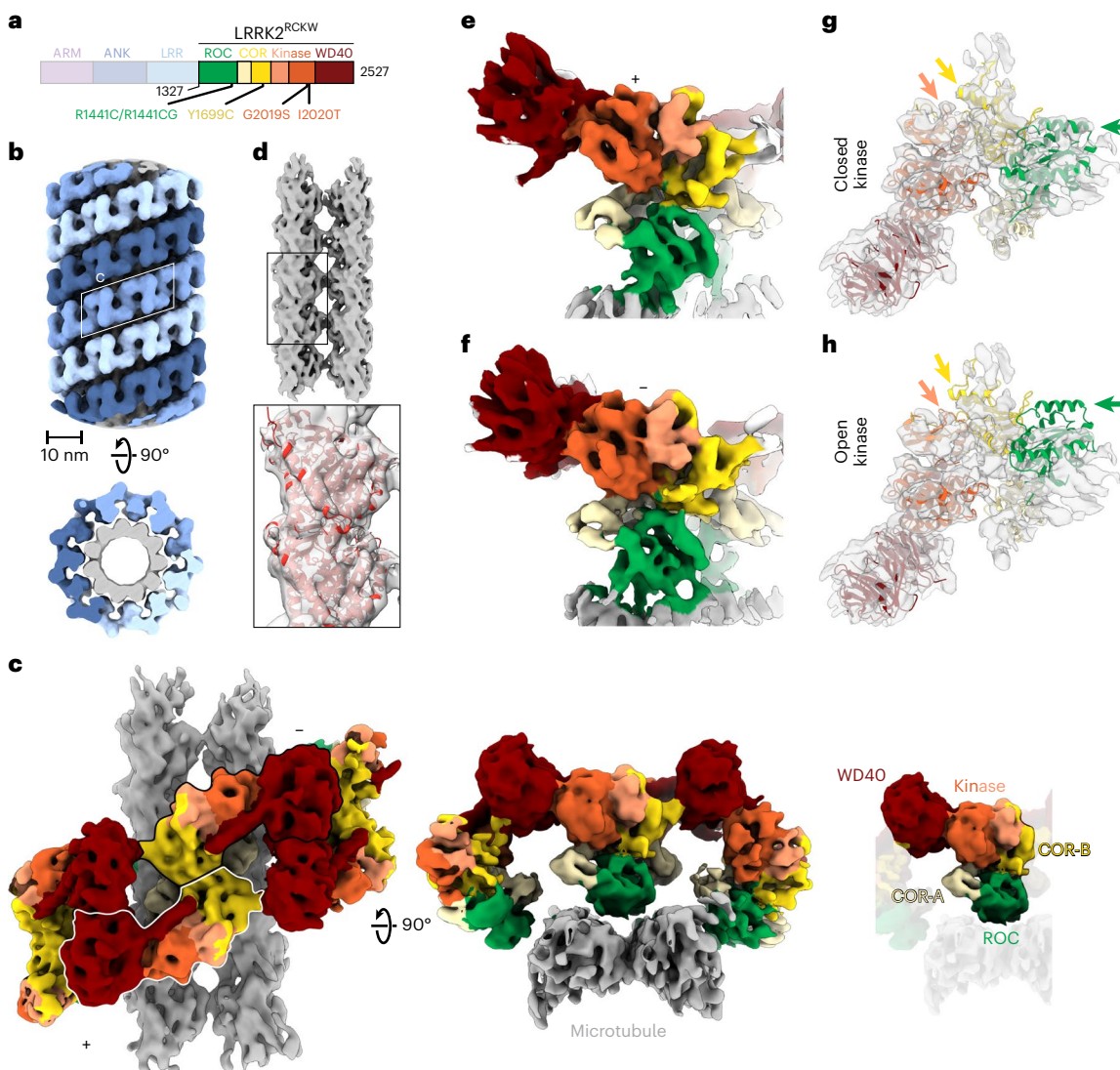

**Fig. 1 | Cryo-EM structure of microtubule-associated LRRK2^RCKW-I2020T.**
**a**, Primary structure of LRRK2. The N-terminal half of LRRK2, absent from the construct used in our cryo-EM studies, is shown in dim colors. The same color-coding of domains is used throughout the figures. **b**, Helical reconstruction (18 Å) of LRRK2^RCKW-I2020T filaments bound to a microtubule in the presence of MLi-2. The three LRRK2^RCKW-I2020T helices are indicated in different shades of blue. **c**, Cryo-EM reconstruction (6.6 Å) of a LRRK2^RCKW tetramer and associated microtubule (two protofilaments), as indicated by the white rhomboid in **b**. Two views are shown, along with a separate representation with a single monomer highlighted and its domains labeled. **d**, Focused refinement of the microtubule in **c** to improve its resolution and determine its polarity. An α/β-tubulin dimer

(from PDB: 6O2R) was docked into the density (black rectangle and inset below). **e,f**, Focused refinement of the '+' (5.0 Å) and '−' (5.2 Å) LRRK2^RCKW-I2020T monomers (as labeled in **c**). **g**, The LRRK2^RCKW domains (ROC, COR-A, COR-B, kinase N-lobe, kinase C-lobe, WD40) (PDB:6VNO) were fitted individually into the 4.5-Å cryo-EM map. **h**, The full LRRK2^RCKW model (PDB: 6VNO) was aligned to the C-lobe of the kinase, as docked in **g**. The colored arrows in **g** and **h** point to parts of the model (PDB:6VNO) that fit the cryo-EM density better when domains are docked individually, allowing the kinase to be in a closed conformation (**g**), but to protrude from it when the full model is used, which has its kinase in an open conformation (**h**).

crucial to determine LRRK2's cellular function and to understand the molecular basis of PD.

LRRK2 is a large, multidomain protein (Fig. 1a). The amino-terminal half contains armadillo, ankyrin, and leucine-rich repeat domains. The carboxy-terminal half contains LRRK2's enzymatic domains—both a Roco family GTPase (Ras-of-complex or ROC domain) and a kinase—as well as a scaffolding domain (C-terminal of ROC, or COR) and a WD40 domain. The COR domain is further subdivided into COR-A and COR-B. Here we refer to the catalytic half of LRRK2 as LRRK2^RCKW, named for its ROC, COR, kinase, and WD40 domains. Recent structures of LRRK2 have revealed the architecture of LRRK2 at near-atomic resolution[35,39]. A 3.5-Å structure of LRRK2^RCKW showed that LRRK2's catalytic half is

J-shaped, placing the kinase and GTPase domains in close proximity[35]. Later, a 3.5-Å structure of full-length LRRK2 revealed that the N-terminal half of LRRK2 wraps around its enzymatic half, with the leucine-rich repeats blocking the kinase's active site in an autoinhibited state[39]. A 14-Å structure of LRRK2 carrying the p.I2020T PD mutation bound to microtubules in cells was obtained using cryo-electron tomography (cryo-ET)[34]. The cryo-ET map was used to guide integrative modeling, leading to a molecular model for the enzymatic half of LRRK2 bound to microtubules[34]. This model was updated when the 3.5-Å cryo-EM structure of LRRK2^RCKW was docked into the cryo-ET structure[35]. In these models, LRRK2^RCKW wraps around the microtubule using two dimerization interfaces, one between WD40 domains and the other

between COR-B domains[35]. In the models, the ROC GTPase domain faces the microtubule, although the cryo-ET structure did not reveal any direct interactions between LRRK2 and the microtubule[34]. An isolated ROC domain has also been shown to interact with alpha- and beta-tubulin heterodimers[40].

We previously investigated the possible functional consequences of LRRK2's interaction with microtubules by looking at the impact of LRRK2 on the movement of the microtubule-based motor proteins dynein and kinesin in vitro[35]. Both dynein and kinesin interact with their membranous cargos directly or indirectly via connections to Rab GTPases, including those Rabs phosphorylated by LRRK2 (refs. [41–44]). Using single-molecule assays, we showed that low nanomolar concentrations of LRRK2[RCKW] blocked the movement of both dynein and kinesin on microtubules[35]. Furthermore, we showed that the conformation of LRRK2's kinase domain was essential for this effect[35]. LRRK2 with its kinase domain 'closed' (the canonical active conformation) by LRRK2-specific type-1 kinase inhibitors blocked motility of dynein and kinesin[35], in agreement with studies showing that these inhibitors enhance the association of LRRK2 with microtubules in cells[33,35,37,38]. In contrast, LRRK2 predicted to have its kinase in an 'open' or inactive conformation (in the presence of type-2 kinase inhibitors) no longer robustly blocked the movement of dynein or kinesin[35].

Despite these insights, how LRRK2 filaments form and interact with microtubules remains unknown. Here, we report cryo-EM structures of microtubule-bound filaments formed by LRRK2[RCKW]. Our structures reveal direct interactions between LRRK2's ROC domain and the microtubule. We show that microtubule binding is mediated by electrostatic interactions and involves the negatively charged C-terminal tubulin tails. We also present a cryo-EM map of the C-terminal half of LRRK1 (LRRK1[RCKW]), LRRK2's closest human homolog. Despite its structural similarity to LRRK2[RCKW], we show that LRRK1[RCKW] does not bind to microtubules. We identify microtubule-facing basic amino acids that are conserved in LRRK2's ROC domain, but not in LRRK1's ROC domain, and are required for LRRK2's interaction with microtubules in vitro and in cells. Mutation of these amino acids also renders LRRK2 unable to block the movement of kinesin in vitro. Together, our work reveals the structural basis for LRRK2's filament formation and microtubule interaction and identifies mutations that perturb them in cells. These are essential advances for determining the cellular functions of LRRK2 and for the further development of therapeutic LRRK2 kinase inhibitors.

## Results

### Cryo-EM structure of microtubule-associated LRRK2[RCKW]

To understand how LRRK2 oligomerizes on and interacts with microtubules, we set out to obtain a higher resolution structure of microtubule-associated LRRK2 filaments using an in vitro reconstituted system and single-particle cryo-EM approaches. We chose to work with LRRK2[RCKW] because it can form filaments in vitro[35] and it accounts for the density observed in the cryo-ET reconstruction of full-length LRRK2 filaments in cells[34].

As has been previously observed[35], co-polymerization of tubulin with LRRK2[RCKW]—either the WT protein, or that carrying the PD-linked mutations p.G2019S or p.I2020T—yielded microtubules decorated with LRRK2[RCKW] (Extended Data Fig. 1a). Diffraction patterns calculated from images of these filaments showed layer lines, indicative of the presence of ordered filaments (Extended Data Fig. 1a). In the presence of MLi-2, a type-1 LRRK2-specific kinase inhibitor[45,46], we saw an additional layer line of lower frequency for all three constructs, suggesting that the filaments had longer-range order (Extended Data Fig. 1a). Unlike WT LRRK2[RCKW] and LRRK2[RCKW]-G2019S, LRRK2[RCKW]-I2020T showed this additional layer line in the absence of MLi-2 as well (Extended Data Fig. 1a). Given these observations, we chose the LRRK2[RCKW]-I2020T filaments that formed in the presence of MLi-2 for our cryo-EM work.

The symmetry mismatch between microtubules, which are polar left-handed helices, and the LRRK2 filaments, which are right-handed and have pseudo-twofold axes of symmetry perpendicular to the microtubule, required that we largely uncouple their processing (Extended Data Fig. 1b,c and Methods). Our cryo-EM analysis resulted in several maps originating from an initial reconstruction of the filaments (Fig. 1b): a map of a LRRK2[RCKW] tetramer that includes density for two microtubule protofilaments (6.6 Å) (Fig. 1c); a higher resolution map of the same LRRK2[RCKW] tetramer excluding the microtubule (5.9 Å) (Extended Data Fig. 3g–i); maps of pseudo-twofold-symmetry-related LRRK2[RCKW] monomers along a filament that face either the plus ('+') (5.0 Å) or minus ('−') (5.2 Å) end of the microtubule, revealing their different contacts with the microtubule (Fig. 1e,f and Extended Data Fig. 1c); and a consensus structure of LRRK2[RCKW] that gives the highest resolution for the kinase domain (4.5 Å) (Extended Data Fig. 1c). The resolutions of our maps, even that of the consensus structure, are not sufficient to reveal how MLi-2 interacts with LRRK2.

The LRRK2[RCKW] filaments are formed by two different homotypic dimer interfaces, involving either COR-B-COR-B or WD40-WD40 interactions (Fig. 1c), in agreement with what modeling has predicted[34,35]. Each interface has a pseudo-twofold axis of symmetry perpendicular to the microtubule axis. The ROC domain points towards and contacts the microtubule (Fig. 1c–f). Our in vitro-reconstituted filaments of LRRK2[RCKW] differ from those formed by full-length LRRK2 in cells[34], with LRRK2[RCKW] forming a triple (rather than double) helix, with the strands packed closer together, likely owing to the absence of the N-terminal half of LRRK2. Despite these differences, the pitch of the helix is similar in both cases (Supplementary Table 1).

We have previously hypothesized that LRRK2's kinase must adopt a closed conformation to form filaments around microtubules[35]. Our current structure agrees with this prediction (Fig. 1g,h). To determine whether the closed conformation of the kinase was a consequence of the presence of MLi-2, which is expected to stabilize that state, we solved a structure of microtubule-associated LRRK2[RCKW] filaments in its absence (Extended Data Fig. 2). Although these filaments are less well ordered (Extended Data Fig. 1a) and thus resulted in a lower resolution reconstruction (7.0 Å), the final map still fit a closed-kinase model of LRRK2[RCKW] better than its open form (Extended Data Fig. 3a,b). Finally, the conformation of the kinase in the microtubule-associated LRRK2[RCKW]-I2020T filaments appears to be more closed than that predicted by AlphaFold[47,48] for the active state of full-length LRRK2 (Extended Data Fig. 3c–f). We cannot determine at this point whether this difference is a consequence of the absence of the N-terminal half of LRRK2, the presence of the p.I2020T mutation in our filaments, a small difference in the AlphaFold modeling, or a consequence of the formation of the filaments themselves.

It has previously been proposed that the ROC domain would mediate binding of LRRK2 to microtubules, owing to its proximity to the microtubule surface in the cryo-ET map of the filaments in cells[34]. However, the cryo-ET map showed no density connecting the ROC domain, or any other domain, to the microtubule[34]. In contrast, our cryo-EM map showed clear density connecting LRRK2[RCKW] and the microtubule (Fig. 1e,f). The fact that microtubules are directional polymers, with '+' (fast-growing) and '−' (slow-growing) ends, means that the ROC domains, which would otherwise be related by a twofold symmetry axis perpendicular to the microtubule, are in different local environments. In agreement with this, their connections to the microtubule became apparent only when LRRK2[RCKW] monomers were refined individually (Fig. 1c,e,f and Extended Data Fig. 1).

### LRRK2's dimer interfaces are important for microtubule association

We next examined the role played by the WD40- and COR-B-mediated dimer interfaces in LRRK2's ability to associate with microtubules. We built a model of the LRRK2[RCKW] filament using rigid-body docking

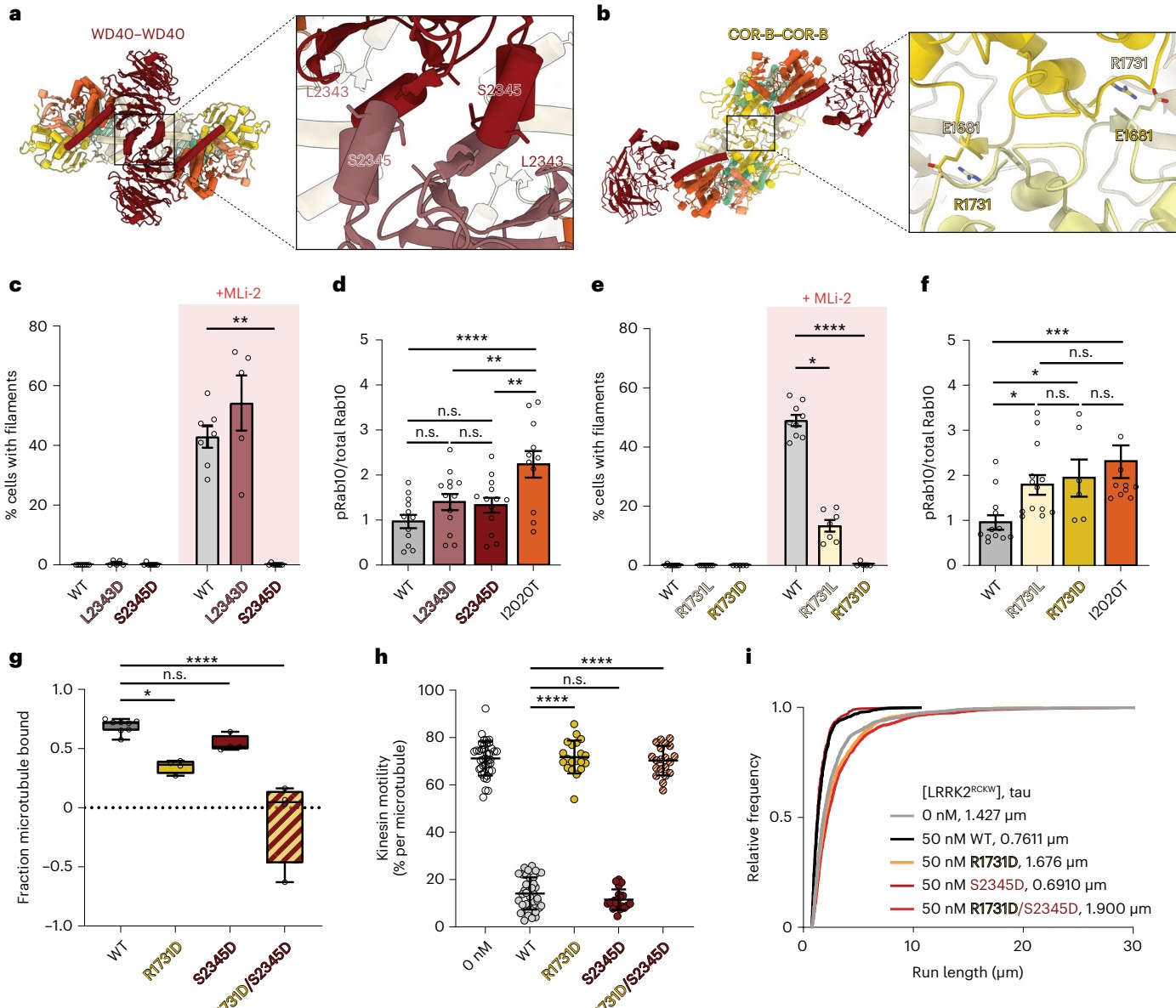

**Fig. 2 | Effect of mutations in LRRK2's WD40 and COR-B domains on filament formation and microtubule binding. a,b**, Dimer interfaces (WD40–WD40 and COR-B–COR-B) involved in filament formation, and the location of the residues tested in this work. **c**, Effect of mutations in the WD40 domain (p.L2343D or p.S2345D) that reduce dimerization of the isolated domain in vitro or the formation of MLi-2-induced filaments in cells. Individual data points represent separate coverslips of cells obtained across at least three independent experiments. Data are mean ± s.e.m. \*\**P* = 0.0076, Kruskal–Wallis test with Dunn's post hoc for multiple comparisons. **d**, Rab10 phosphorylation in 293T cells overexpressing WT LRRK2, LRRK2 carrying mutations in the WD40 domain, or LRRK2-I2020T, which increases Rab10 phosphorylation in cells. 293T cells were transiently co-transfected with plasmids encoding GFP-LRRK2 (WT or mutant) and GFP-Rab10. Quantified immunoblotting data are shown as p-Rab10/total GFP-Rab10 ratios, normalized to the average of all WT values. Individual data points represent separate populations of cells obtained across at least three independent experiments. Data are mean ± s.e.m. \*\*\*\**P* < 0.0001, \*\**P* < 0.0052, one-way ANOVA followed by a Fisher's least-significant difference test. **e**, Effect of mutations (p.R1731L or p.R1731D) at the COR-B–COR-B interface on the formation of MLi-2-induced filaments in cells. Individual data points represent separate coverslips of cells obtained from at least three independent experiments. Data are mean ± s.e.m. \*erP* = 0.0205, \*\*\*\**P* < 0.0001, Kruskal–Wallis test with Dunn's post hoc for multiple comparisons. **f**, Rab10 phosphorylation

in 293T cells overexpressing WT LRRK2, LRRK2 with mutations in the COR-B domain, or LRRK2-I2020T. 293T cells were treated as in **d**. Data are quantified and shown as in **d**. Data are mean ± s.e.m. \**P* < 0.035, \*\*\**P* = 0.0010, one-way ANOVA followed by a Fisher's least-significant difference test. **g**, Effect of mutations in the WD40 or WD40 and COR-B domains on the binding of LRRK2^RCKW to microtubules in a microtubule pelleting assay. Box and whisker plot center line denotes the median value; whiskers denote the minimum and maximum values. \**P* = 0.0111, \*\*\*\**P* < 0.0001, one-way ANOVA with Dunnett's multiple comparisons test. WT, *n* = 8 replicates; mutants, *n* = 4 replicates. **h**, Effect of mutations in the WD40 or WD40 and COR-B domains on the inhibition of kinesin motility in vitro by 50 nM LRRK2^RCKW. Inhibition of kinesin motility was quantified as percentage of motile events per microtubule. Data points represent individual microtubules obtained across at least two independent experiments. Data are mean ± s.d. \*\*\*\**P* < 0.0001, Kruskal–Wallis test with Dunn's post hoc for multiple comparisons. **i**, Cumulative distribution of run lengths for kinesin in the absence or presence of 50 nM LRRK2^RCKW (WT or carrying WD40, COR-B, or WD40 and COR-B mutations). The run lengths were not significantly different between 50 nM WT and LRRK2^RCKW-S2345D, and were significantly different between 50 nM WT LRRK2^RCKW and LRRK2^RCKW-R1731D S2345D and LRRK2^RCKW-R1731D (Kruskal–Wallis test with Dunn's post hoc for multiple comparisons). Mean decay constants (tau) are shown.

of individual domains from the LRRK2[RCKW] structure (PDB: 6VNO)[35] (Fig. 2a,b). This revealed WD40–WD40 and COR-B–COR-B interfaces that are very similar to those seen previously with isolated WD40 domains[49], full-length LRRK2 COR-B–COR-B dimers[39], and LRRK2[RCKW] dimers in the absence of microtubules[35]. However, small differences exist between our model and the full-length LRRK2 dimer[39] (Extended Data Fig. 3g–j). It remains to be seen whether these differences are due to the absence of the N-terminal half of LRRK2 in the microtubule-associated filaments or to small conformational changes associated with filament formation.

On the basis of our model, we made mutants designed to disrupt both interfaces and tested their ability to form filaments in cells and to bind microtubules or inhibit the motility of kinesin in vitro. At the WD40–WD40 interface, we mutated leucine 2343 or serine 2345 to aspartic acid (p.L2343D or p.S2345D; Fig. 2a), designed to introduce a charge clash. At the COR-B–COR-B dimer interface, we mutated arginine 1731 to leucine or aspartic acid (p.R1731L or p.R1731D; Fig. 2b), designed to disrupt the salt bridge with glutamic acid 1681. The expression levels of all mutant alleles were similar to that of full-length WT LRRK2 when transfected into 293T cells (Extended Data Fig. 4a–d). We then tested the ability of these mutations to disrupt filament formation by LRRK2 (WT except for the interface mutations) in cells, which is induced by MLi-2 (refs. [33,35,37,38]) (Fig. 2c,e and Extended Data Fig. 4a,b). As has previously been shown, mutation of either the WD40–WD40 interface[34] or the COR-B–COR-B interface[39] reduced filament formation in cells. We found that p.S2345D, p.R1731L, and p.R1731D all significantly decreased MLi-2-induced LRRK2 filament formation, with p.S2345D and p.R1731D completely abolishing our ability to detect filaments in cells (Fig. 2c,e). Surprisingly, although the p.L2343D mutation has previously been shown to decrease dimerization of a purified WD40 domain[49], it did not reduce the formation of LRRK2 filaments in the presence of MLi-2 (Fig. 2c). We also tested each mutant's ability to phosphorylate Rab10 in cells, and found that the WD40 dimerization interface mutants had no effect on LRRK2's kinase activity, whereas the COR-B dimerization interface mutants roughly doubled it (Fig. 2d,f, and Extended Data Fig. 4c,d).

Next, we examined the effects of the mutations at the LRRK2 dimerization interfaces on LRRK2's ability to bind microtubules or inhibit kinesin motility in vitro. To investigate LRRK2's binding to microtubules, we incubated pure LRRK2[RCKW] with in vitro-assembled, taxol-stabilized microtubules and quantified the fraction of LRRK2 that pelleted with microtubules after centrifugation. Although a point mutation at the WD40 dimerization interface (p.S2345D) did not affect LRRK2[RCKW]'s pelleting with microtubules, a point mutation at the COR-B interface (p.R1731D) reduced it by about 50% (Fig. 2g and Extended Data Fig. 4e). Combining these mutations (p.R1731D and p.S2345D) largely abolished LRRK2[RCKW]'s interaction with microtubules (Fig. 2g and Extended Data Fig. 4e). Cryo-EM imaging of microtubules incubated with the different mutants agreed with the binding data: we observed the layer lines that were indicative of filament formation with LRRK2[RCKW]-S2345D, but not with LRRK2[RCKW]-R1731D or LRRK2[RCKW]-R1731D S2345D (Extended Data Fig. 4f). Previously, we showed that low nanomolar concentrations of LRRK2[RCKW] blocked the movement of dynein and kinesin in vitro[35]. To determine whether the dimerization interfaces are required for this inhibitory effect, we monitored the motility of single GFP-tagged human kinesin-1 (referred to as 'kinesin' here) molecules using total internal reflection fluorescence (TIRF) microscopy. As in the microtubule-binding experiments, we found that LRRK2[RCKW]-S2345D blocked kinesin motility similarly to WT LRRK2[RCKW], although the COR-B interface mutant LRRK2[RCKW]-R1731D or the combined interface mutant LRRK2[RCKW]-R1731D S2345D no longer inhibited kinesin motility in vitro (Fig. 2h,i and Extended Data Fig. 4g). Importantly, 2D averages from cryo-EM images of LRRK2[RCKW]-R1731D S2345D showed that the mutations do not alter the structure of the protein substantially (Extended Data Fig. 4h).

## Electrostatic interactions drive binding of LRRK2[RCKW] to microtubules

We next tested the hypothesis that LRRK2 binding to microtubules is mediated by electrostatic interactions between the negatively charged surface of the microtubule and basic residues in LRRK2's ROC domain. In addition to the observed charge complementarity between our model of the LRRK2 filaments and the microtubule (Fig. 3a and Deniston et al.[35]), other data support this hypothesis: (1) the symmetry mismatch between microtubules and the LRRK2 filaments suggests that there cannot be a single LRRK2-microtubule interface[34], (2) the cryo-ET reconstruction of filaments in cells showed no clear direct contact between LRRK2 and tubulin[34], and (3) the connections in our reconstruction only became apparent when LRRK2[RCKW] monomers were refined individually (Fig. 1c,e,f and Extended Data Fig. 1). To directly test this hypothesis, we developed a fluorescence-based assay to monitor binding of LRRK2[RCKW] to microtubules in vitro. We randomly chemically labeled primary amines of LRRK2[RCKW] with BODIPY TMR-X ('TMR' here) and used widefield fluorescence microscopy to quantify the association of TMR-LRRK2[RCKW] with Alexa Fluor 488-labeled microtubules tethered to a coverslip. Chemical labeling did not significantly impair LRRK2[RCKW] kinase activity as assessed by Rab8a phosphorylation in vitro (Extended Data Fig. 4i,j). In our indirect assay of filament formation, TMR-LRRK2[RCKW] also inhibited the microtubule-based motility of kinesin (Extended Data Fig. 4k), suggesting that its ability to bind microtubules was not compromised. Titration of increasing concentrations of TMR-LRRK2[RCKW] to microtubules led to a dose-dependent increase in microtubule binding (Fig. 3b,c). Notably, LRRK2[RCKW] bound to microtubules at low nanomolar concentrations, similar to the concentrations required to inhibit the motility of kinesin and dynein[35]. Unlabeled LRRK2[RCKW] also bound microtubules in a bulk microtubule co-sedimentation assay (Extended Data Fig. 4l,m).

To determine whether electrostatic interactions contribute to the binding of LRRK2[RCKW] to microtubules, we tested the effect of increasing concentrations of sodium chloride on this binding. We observed a dose-dependent decrease in microtubule binding (Fig. 3d and Extended Data Fig. 4n). We also observed a salt-dependent decrease in microtubule binding for unlabeled LRRK2[RCKW] in the bulk co-sedimentation assay (Extended Data Fig. 4l,m).

Tubulin carries an overall negative charge, and the disordered, negatively charged, glutamate-rich C-terminal tails of tubulin are known to contribute to microtubule binding by many microtubule-associated proteins[50]. We tested the contribution of the tubulin tails to the LRRK2[RCKW]-microtubule interaction by removing them with the protease subtilisin, which cleaves tubulin near its C terminus[51] (Extended Data Fig. 4o). Cleavage of tubulin tails decreased LRRK2[RCKW]'s ability to bind microtubules by ~50% (Fig. 3e and Extended Data Fig. 4p). Together, these results show that LRRK2's interaction with the microtubule is driven by electrostatic interactions and is mediated in part by the C-terminal tails of tubulin.

## LRRK1[RCKW] adopts a similar overall fold to LRRK2[RCKW]

To identify specific residues in LRRK2 that might be important for mediating its interaction with microtubules, we used a comparative approach with its closest homolog, LRRK1. Although LRRK2 has been linked to both familial and sporadic PD[1–6], LRRK1 is not clinically associated with PD[52], but instead is implicated in metabolic bone disease and osteopetrosis[53–56]. Many of LRRK1's domains are relatively well conserved with LRRK2, with 41%, 48%, 46%, and 50% similarity between the leucine-rich repeat (LRR), ROC, COR, and kinase domains, respectively. The N and C termini of LRRK1 and LRRK2 are more divergent; LRRK1 lacks the N-terminal armadillo repeats, and it was unclear at the time (this part of our work was done before the release of Alpha-Fold[47,48]), on the basis of sequence analyses, whether LRRK1, like LRRK2, contained a WD40 domain, with only 27% sequence similarity in this region.

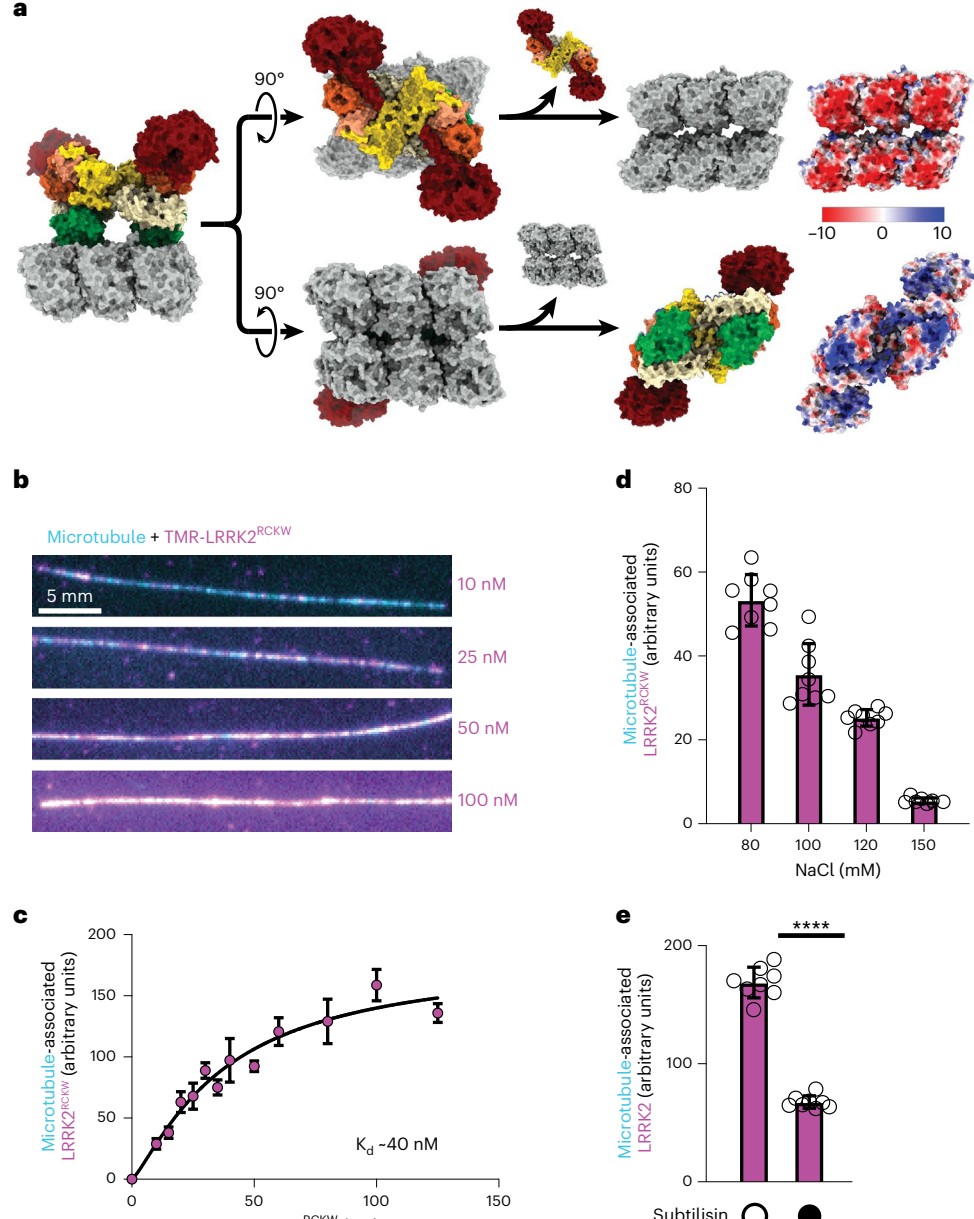

**Fig. 3 | LRRK2^RCKW interacts with the microtubule via electrostatic interactions.**
**a**, Charge distribution in the molecular model for microtubule-associated LRRK2^RCKW filaments (Fig. 1). The model is shown in surface representation on the left and is then split to reveal the microtubule surface facing LRRK2^RCKW (top) or the LRRK2^RCKW surface facing the microtubule (bottom). The Coulomb potential of those surfaces is shown on the right. The acidic C-terminal tubulin tails that further contribute negative charge density to the microtubule are disordered in our structure and are not included here. **b**, Representative images of randomly labeled TMR-LRRK2^RCKW (magenta), bound to microtubules labeled with Alexa Fluor 488 and tethered to a coverslip (cyan). The concentrations of TMR-LRRK2^RCKW

are indicated on the right. **c**, Quantification of data represented in **b**. Images were flatfield corrected, average TMR fluorescence intensity was measured along each microtubule in each field of view, and an average value per field of view was calculated, normalized for microtubule length. Data are mean ± s.d., $n = 8$ fields of view. **d**, Binding of 100 nM TMR-LRRK2^RCKW to microtubules in the presence of increasing concentrations of sodium chloride, quantified from the assay exemplified by **b**. **e**, Binding of 50 nM TMR-LRRK2^RCKW to microtubules untreated or pre-treated with subtilisin, quantified from the assay exemplified by **b**. Data are mean ± s.d., $n = 8$ fields of view. ****$P < 0.0001$, unpaired two-tailed $t$-test with Welch's correction.

We began by solving a cryo-EM structure of the part of LRRK1 that corresponds to LRRK2^RCKW (residues 631 to 2015; referred to as LRRK1^RCKW) (Fig. 4a and Extended Data Fig. 5a). The resolution of the LRRK1^RCKW monomer (5.8 Å) was limited by the fact that the protein adopted the same strong preferred orientation that we had observed for LRRK2^RCKW (ref. 35). Although LRRK2^RCKW forms trimers, which allowed us to solve its high-resolution structure[35], we saw no evidence of trimer formation by LRRK1^RCKW. Our structure, obtained in the presence

of GDP but in the absence of ATP, shows that LRRK1^RCKW adopts the same overall J-shaped domain organization seen in LRRK2^RCKW and contains a WD40 domain (Fig. 4a,b). Our map revealed that the αC helix in the N-lobe of LRRK1's kinase is about four turns longer than that in LRRK2 (Fig. 4c), a feature correctly predicted by the AlphaFold[47,48] model of LRRK1. Our structure also revealed a density corresponding to a C-terminal helix extending from the WD40 domain and lining the back of the kinase domain, as is the case for LRRK2, but the LRRK1 helix

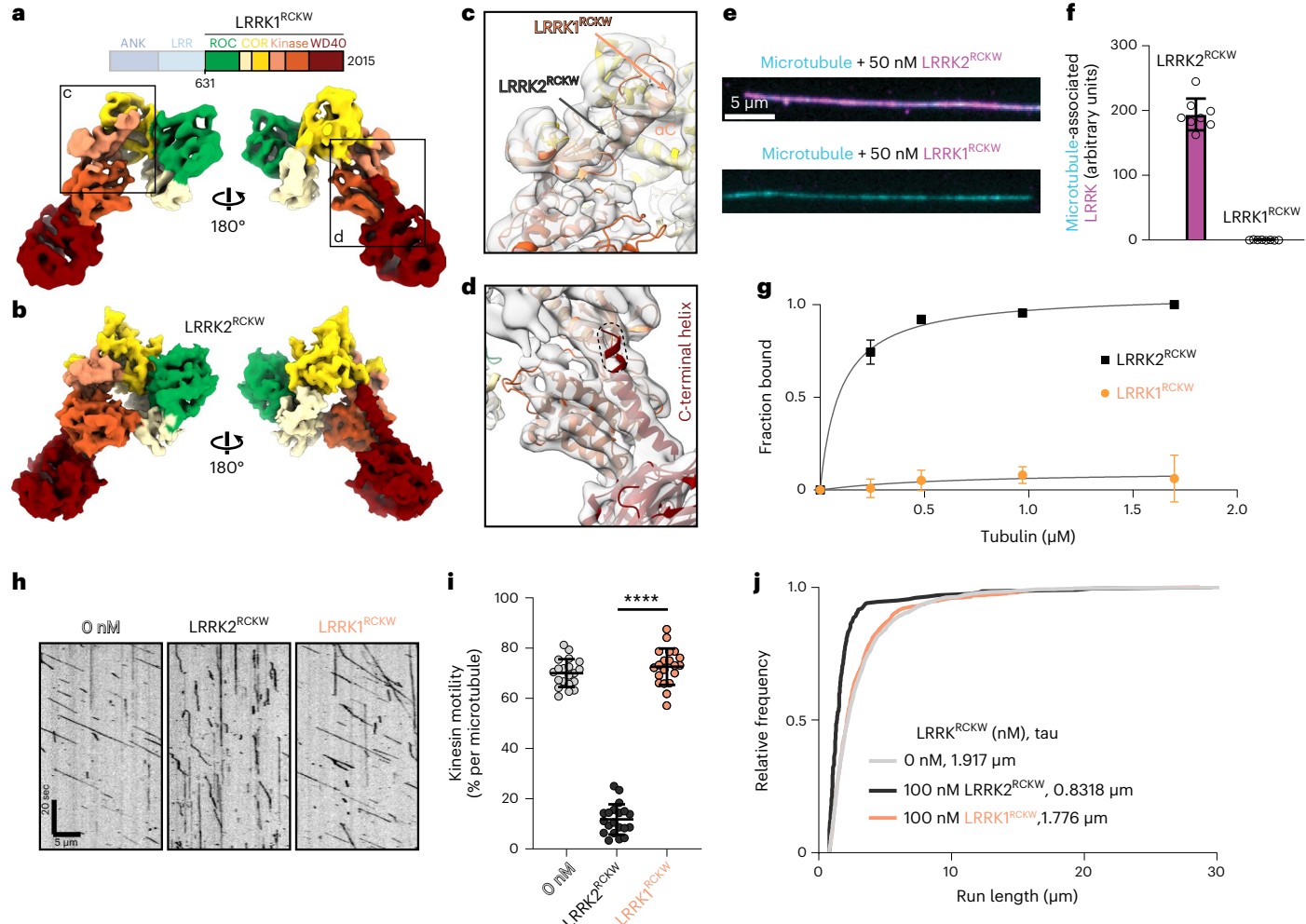

**Fig. 4 | LRRK1^RCKW is structurally similar to LRRK2^RCKW but does not bind to microtubules. a**, Cryo-EM map (5.8 Å) of a LRRK1^RCKW monomer, with domains colored according to the scheme shown above. **b**, The molecular model for LRRK2^RCKW (PDB: 6VNO) is shown as a calculated 6-Å density (molmap command in ChimeraX), in the same orientations used for LRRK1^RCKW in **a**. **c,d**, Close-ups of the LRRK1^RCKW map shown in **a**, with the AlphaFold model of LRRK1 docked into it. These close-ups highlight the difference in length in the αC helix between LRRK1 and LRRK2 (**c**), and a difference at the C-terminal helix emerging from the WD40 domain between our experimental map of LRRK1^RCKW and the AlphaFold model of LRRK1 (**d**). **e**, Representative images of Alexa Fluor 488-labeled microtubules (cyan) incubated with 50 nM of either LRRK2^RCKW (magenta, top) or LRRK1^RCKW (magenta, bottom). **f**, Quantification of data in **e**, as outlined in Figure 3c. Data

are mean ± s.d., n = 8 fields of view. **g**, Microtubule pelleting assay for 200 nM LRRK2^RCKW or LRRK1^RCKW with increasing tubulin concentrations. Data are mean ± s.d., n = 4. The solid lines represent a hyperbolic curve fit to the data. **h**, Example kymographs for single-molecule kinesin motility assays alone or in the presence of 100 nM of either LRRK2^RCKW or LRRK1^RCKW. **i**, Quantification of data in **h** as percentage of motile kinesin events per microtubule. Data are mean ± s.d. ****P < 0.0001, Kruskal–Wallis test with Dunn's post hoc for multiple comparisons. **j**, Cumulative distribution of run lengths for kinesin in the absence or presence of 100 nM LRRK2^RCKW or LRRK1^RCKW. The run lengths were not significantly different between 0 nM and 100 nM LRRK1^RCKW conditions (Kruskal–Wallis test with Dunn's post hoc for multiple comparisons). Data are from two biological replicates, with three or two technical replicates of each experiment.

appears to be shorter (Fig. 4d). This disagrees with the LRRK1 structure predicted by AlphaFold, which has a longer C-terminal helix (Fig. 4d). The meaning of this difference is not clear at this time, as the AlphaFold structure was modeled in the active conformation (closed kinase), whereas our LRRK1^RCKW is in an inactive, open-kinase conformation and lacks the amino-terminal repeats. At the current resolution, LRRK2^RCKW and LRRK1^RCKW are otherwise very similar, confirming that the overall domain organization is conserved between these two proteins.

## LRRK1^RCKW does not bind microtubules

Given the structural similarity between LRRK1 and LRRK2 (Fig. 4a,b), we wondered whether LRRK1 could also bind microtubules. To test this, we randomly chemically labeled LRRK1^RCKW with BODIPY TMR-X and used widefield fluorescence microscopy to quantify microtubule binding in vitro. We did not observe association of TMR-LRRK1^RCKW with

microtubules (Fig. 4e,f). Unlabeled LRRK1^RCKW also failed to co-sediment with microtubules (Fig. 4g and Extended Data Fig. 5b) or block kinesin motility, even at a concentration of 100 nM (Fig. 4h–j). Together, these data show that, in contrast to LRRK2, LRRK1 does not interact with microtubules.

## Basic residues in LRRK2's ROC domain are important for microtubule binding

Next, we used our discovery that LRRK1^RCKW and LRRK2^RCKW share a similar structure, but only LRRK2^RCKW binds microtubules, to identify amino acids in LRRK2 that are important for microtubule binding. A sequence alignment of LRRK1 and LRRK2 revealed several basic patches in LRRK2's ROC domain that are not well conserved in LRRK1 (Extended Data Fig. 5c). These basic patches create a positively charged surface facing the microtubule that is absent in LRRK1 (Fig. 5a,b).

The patches correspond to residues 1356–1359 (KTKK in human LRRK2), 1383–1386 (KRKR in human LRRK2), and 1499–1502 (KLRK in human LRRK2). In our highest resolution maps, in which we refined individual LRRK2$^{RCKW}$ monomers in the filament and their contacts with the microtubule (Fig. 1e,f), the strongest density connecting LRRK2$^{RCKW}$ to tubulin involves the 1356–1359 and 1383–1386 basic patches (Fig. 5c and Extended Data Fig. 5d). To determine whether these basic patches are required for LRRK2's interaction with microtubules, we mutated two basic residues to alanine in each patch in the context of LRRK2$^{RCKW}$ (p.K1358A and p.K1359A or p.R1384A and p.K1385A) and tested the the ability of each mutant to bind to microtubules in vitro. Both mutants showed a significant decrease in microtubule binding in a microtubule co-sedimentation assay compared with WT LRRK2$^{RCKW}$ (Fig. 5d). LRRK2$^{RCKW}$-K1358A K1359A also showed a significant reduction in its inhibition of kinesin motility in vitro compared with WT LRRK2$^{RCKW}$ (Fig. 5e,f and Extended Data Fig. 5e). Finally, we introduced full-length GFP-LRRK2, carrying either of the two basic-patch mutations, into human 293T cells and quantified microtubule association in the absence or presence of MLi-2. In the absence of MLi-2, all three constructs (WT and the two basic-patch mutants) formed little or no filaments in cells (Fig. 5g and Extended Data Fig. 5f). Treatment with MLi-2 resulted in the appearance of filaments in a significant percentage of cells carrying WT LRRK2, but failed to induce filament formation in cells carrying the basic-patch mutants (Fig. 5g and Extended Data Fig. 5f). We also tested whether GFP-LRRK2 carrying the PD-linked p.I2020T mutation, which forms filaments in cells in the absence of MLi-2 (refs. 33,35,37,38) (Fig. 5h), is sensitive to a basic-patch mutation. Indeed, GFP-LRRK2-I2020T no longer formed microtubule-associated filaments in cells while carrying the p.K1358A and p.K1359A mutations (Fig. 5h and Extended Data Fig. 5g). Cryo-EM imaging of microtubules incubated with LRRK2$^{RCKW}$ carrying either of the two basic-patch mutants did not show the layer lines that are indicative of filament formation (Extended Data Fig. 4f). Class averages from cryo-EM images of the soluble form of those mutants also showed that the mutations do not alter the structure of the protein substantially (Extended Data Fig. 4h).

Although none of the most common PD-linked mutations in LRRK2 are found in these basic-patch regions, the recently reported p.R1501W variant[57] is found in the ROC domain facing the microtubule, near the basic patches we identified (Fig. 5c). To determine whether p.R1501W alters LRRK2's interaction with microtubules, we expressed GFP-LRRK2-R1501W in 293T cells. In the presence of MLi-2, LRRK2-R1501W showed a ~50% reduction in the fraction of cells containing microtubule-bound filaments compared with WT LRRK2 (Fig. 5i and Extended Data Fig. 5h). Although the effect of the p.R1501W mutation

was milder than that of the basic-patch mutations in the context of WT LRRK2, it was as extreme as the basic-patch mutants when combined with the p.I2020T mutation, where it also abolished filament formation (Fig. 5j and Extended Data Fig. 5g).

Importantly, none of the effects described above are due to changes in protein expression levels (Extended Data Fig. 5i) or to changes in the kinase activity of LRRK2 (Extended Data Fig. 5i,j). LRRK2 with the basic-patch mutants and LRRK2-R1501W show similar levels of Rab10 phosphorylation in cells compared with WT LRRK2, and they do not alter the increased Rab10 phosphorylation seen in the context of p.I2020T (Extended Data Fig. 5i,j).

## Discussion

Here we report a structure of in vitro reconstituted LRRK2$^{RCKW}$ filaments bound to microtubules. Our structure confirmed our previous proposal that filament formation by LRRK2 requires its kinase to be in a closed (active) conformation[35]. This provides a structural explanation for the observation that LRRK2-specific type-1 kinase inhibitors, which are expected to stabilize the closed conformation, induce filament formation in cells[34,35,37,38]. We also report a structure of the catalytic half of LRRK1 (LRRK1$^{RCKW}$), LRRK2's closest homolog, which shows that the overall fold is similar in both LRRK proteins. Despite this similarity, LRRK1 does not bind to microtubules in vitro. We identified microtubule-facing basic patches in LRRK2's ROC domain that are not well conserved in LRRK1 and these are located in regions of the cryo-EM map showing density connecting LRRK2$^{RCKW}$ to the microtubule. Mutating two basic amino acids in LRRK2's ROC domain was sufficient to block microtubule binding both in cells and in vitro. Together, the results of this work provide important insights and tools for probing the cellular function and localization of LRRK2 and for designing LRRK2-specific kinase inhibitors.

The previous reconstruction of LRRK2 filaments in cells, at 14 Å, did not show any density connecting LRRK2 to the microtubule. The higher resolution of our map, and our ability to process LRRK2$^{RCKW}$ monomers individually, allowed us to circumvent the symmetry mismatch between the LRRK2 filaments and the microtubule, and show that LRRK2$^{RCKW}$ monomers related by (pseudo) twofold symmetry in the filament are indeed not truly symmetric and interact with the microtubule differently. The general features of the filaments—their curvature and basic patches facing a negatively charged surface (the microtubule)—raise the possibility that a similar geometry could be involved in the interaction between LRRK2 and membranes.

Although LRRK2 filaments were double-helical in cells[34], the in vitro reconstituted LRRK2$^{RCKW}$ filaments were triple-helical and packed closer together. However, the helical parameters are very

**Fig. 5 | Basic patches in the ROC domain are involved in LRRK2's binding to microtubules. a,b**, Surface charge distribution (Coulomb potential) for LRRK2$^{RCKW}$ (PDB: 6VNO) (**a**) and the AlphaFold model for LRRK1$^{RCKW}$ (**b**). The green oval on the right highlights the region in the ROC domain facing the microtubule in the filament structure where basic patches are present (and conserved) in LRRK2 but absent in LRRK1. **c**, Molecular model of the microtubule-bound LRRK2$^{RCKW}$ filament with tubulin, shown in surface representation. '+' and '−' indicate the two monomers in a dimer. Close-ups, shown as insets labeled (i) and (ii) below the structures, highlight basic residues near the microtubule surface tested here. **d**, Binding of LRRK2$^{RCKW}$, either WT or carrying mutations in the ROC domain's basic patches, to microtubules using a microtubule pelleting assay. Box and whisker plot center line denotes the median value, and whiskers denote minimum and maximum values. ***P = 0.0006, ****P < 0.0001, one-way ANOVA with Dunnett's multiple comparisons test. n = 4 replicates. **e**, Single-molecule motility assays for kinesin alone or in the presence of increasing concentrations of either WT LRRK2$^{RCKW}$ or LRRK2$^{RCKW}$ carrying mutations in the ROC domain. Inhibition of kinesin motility was quantified as percentage of motile events per microtubule. Data points represent individual microtubules obtained across at least two independent experiments. Data are mean ± s.d. ***P < 0.0002,

****P < 0.0001, Kruskal–Wallis test with Dunn's post hoc for multiple comparisons. **f**, Cumulative distribution of run lengths for kinesin in the absence or presence of 100 nM LRRK2$^{RCKW}$ (WT or carrying ROC mutations). Run lengths were significantly different between 100 nM WT LRRK2$^{RCKW}$ and LRRK2$^{RCKW}$-K1358A K1359A (Kruskal–Wallis test with Dunn's post hoc for multiple comparisons). Mean decay constants (tau) are shown. **g**, Quantification of microtubule-associated filament formation in cells expressing WT GFP-LRRK2, GFP-LRRK2-K1358A K1359A, or GFP-LRRK2-R1384A K1385A in the absence or presence of MLi-2. Data are mean ± s.e.m. **P = 0.0022, ****P < 0.0001, Kruskal–Wallis test with Dunn's post hoc for multiple comparisons. **h**, Quantification of microtubule-associated filament formation in cells expressing GFP-LRRK2-I2020T or GFP-LRRK2-K1358A K1359A I2020T. Data are mean ± s.e.m. ****P < 0.0001, two-tailed Mann–Whitney test. **i**, Same as for **g** for a recently identified PD-linked mutation in the ROC domain (p.R1501W). Data are mean ± s.e.m. **P = 0.0017, two-tailed Mann–Whitney test. **j**, Same as for **h** for GFP-LRRK2-I2020T and GFP-LRRK2-R1501W I2020T. Data are mean ± s.e.m. ***P = 0.0002, two-tailed Mann–Whitney test. Individual data points in **g–j** represent separate coverslips of cells obtained across at least four independent experiments.

similar between the structures, suggesting that the underlying structure of the filaments is similar as well. The most likely explanation for the differences is the absence of the N-terminal repeats in our structure of the LRRK2^RCKW filaments. Although the N-terminal half of LRRK2 was present in the filaments reconstructed in cells, they were disordered and absent from the final map[34]. Placing the AlphaFold model of LRRK2 into the cryo-ET map of filaments in cells showed major clashes between the filament itself (formed by the RCKW domains) and the N-terminal repeats (Extended Data Fig. 6). Thus, the filaments could not form unless the N-terminal repeats are undocked from the rest of the protein. Their presence, albeit in a flexible state, could explain the

larger spacing, and thus lower number of helices, seen in LRRK2 versus LRRK2^RCKW filaments; the disordered N-terminal repeats could act as 'spacers' that prevent the filaments from packing closer together.

Our data suggest that LRRK2 can bind microtubules as very short oligomers. This binding mode is likely to be the preponderant one at the low concentrations used in our in vitro single-molecule assays. We base these observations on the mutants we designed to disrupt dimerization interfaces (COR-B and WD40). Any mutant that completely abolishes one dimerization interface would allow LRRK2 to form dimers (via the other interface) but would prevent the formation of longer oligomers. Although mutants predicted to break either the COR-B (p.R1731D) or

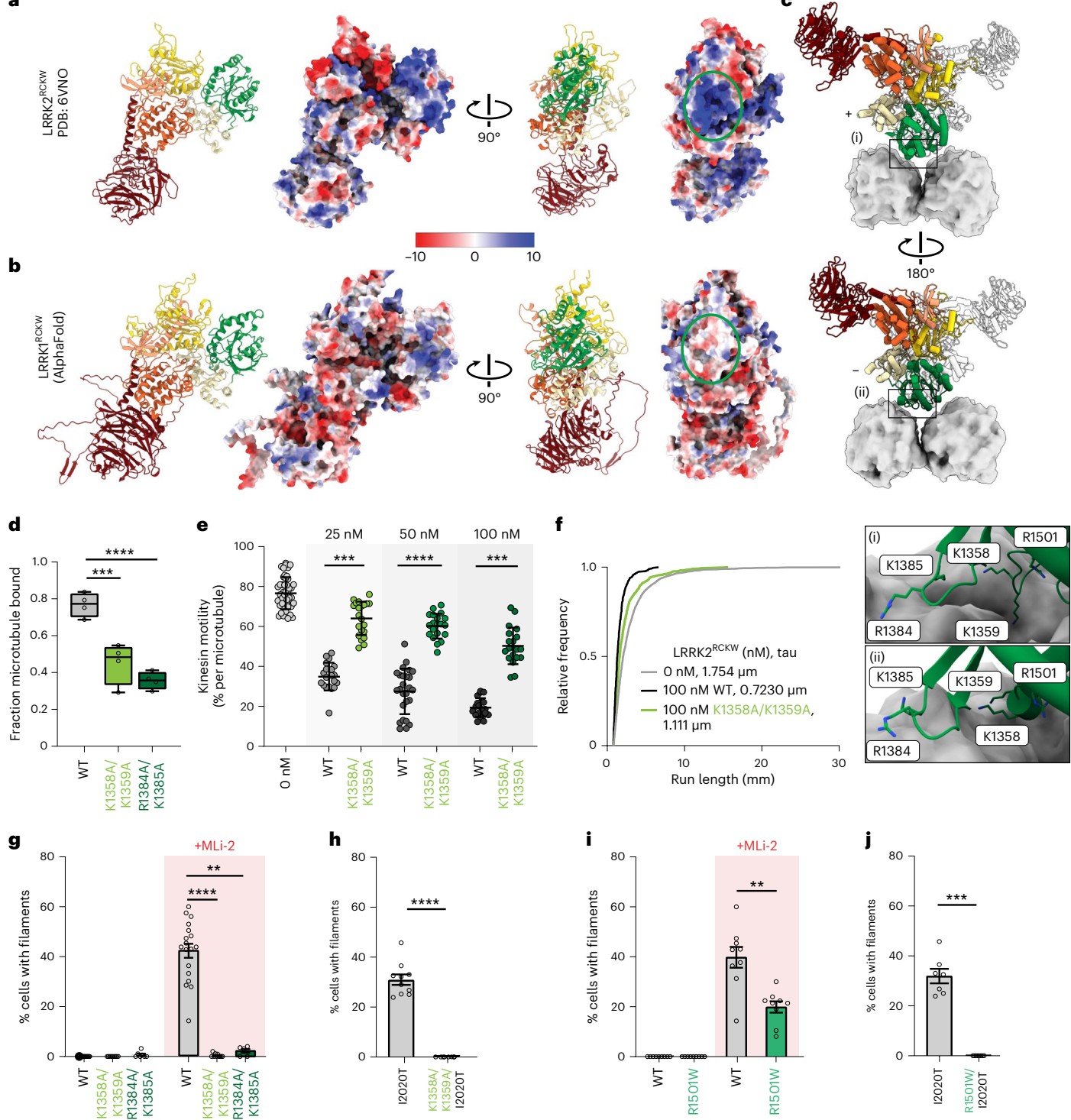

WD40 (p.S2345D) interfaces abolished formation of LRRK2 filaments in cells, their effects on microtubule binding in vitro were far less extreme, with p.R1731D resulting in a ~50% decrease and p.S2345D having no significant effect. The p.S2345D mutant likely does not fully disrupt the WD40 interface, as it is able to form some filaments in vitro under the high concentrations used for cryo-EM, although these filaments are less well ordered. The mutants' affinity for microtubules correlates with their ability to inhibit kinesin motility: p.R1731D is unable to inhibit the motor, whereas p.S2345D inhibits motility as much as WT LRRK2 does. Taken together, these data suggest that small LRRK2 oligomers, as small as a dimer, could act as roadblocks for microtubule-based transport. This possibility, along with the fact that type-1 inhibitors stabilize the conformation of LRRK2 that favors microtubule binding, should be considered when designing LRRK2 inhibitors.

An intriguing observation from our data is the increase in Rab10 phosphorylation in cells expressing LRRK2 with the p.R1731D mutation, designed to disrupt the COR-B interface. Since the available structural information shows that LRRK2 adopts the same autoinhibited conformation in its monomeric and dimeric forms, this was not a result we had predicted. One possible explanation is that conformational changes involved in the activation of LRRK2 are favored in its monomeric form, which the p.R1731D mutation would promote. Alternatively, this effect could reflect differences in cellular localization between the monomer and the dimer, which could in turn change their exposure to the Rab10 substrate.

Although LRRK2 readily binds microtubules at low concentrations in vitro, whether LRRK2 binds to and/or forms filaments around microtubules in cells expressing endogenous levels of LRRK2 remains an open question. Although the only reports of LRRK2 interacting with microtubules in cells so far have been under overexpression conditions[18,33–35,38], only a limited number of cell types have been imaged for LRRK2 localization, and to our knowledge there are no reports of live-cell imaging of endogenous LRRK2. Thus, an important future goal will be to determine the localization and dynamics of LRRK2 expressed at endogenous levels in PD-relevant cell types. A recent report suggests that a noncoding *LRRK2* PD variant leads to increased *LRRK2* expression in induced microglia[58]. In addition, *LRRK2* expression levels are elevated in a variety of immune cells in people with PD compared with age-matched healthy controls[59,60]. These findings raise the possibility that increased expression of WT *LRRK2* could be linked to PD. Our finding that the interaction of WT LRRK2[RCKW] with microtubules acts as a potent roadblock for the microtubule-based motors dynein and kinesin[35] suggests a mechanism for how increased *LRRK2* expression levels could be detrimental for membrane trafficking. All of the membrane cargos that LRRK2 has been implicated in trafficking—including lysosomes, endo-lysosomes, autophagosomes, and mitochondria[14]—are moved by dynein and kinesin[61–63]. Elevated LRRK2 kinase activity leading to the phosphorylation of Rab GTPases is also linked to changes in membrane trafficking, and specifically in the recruitment of adapter proteins that can bind dynein and kinesin motors[64,65]. Thus, examining the effects of increased *LRRK2* expression in combination with increased LRRK2 kinase activity may be relevant for understanding the molecular basis of PD.

## Online content

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

## Methods

### Cloning, plasmid construction, and mutagenesis

LRRK2[RCKW] and Rab8a protein expression vectors were cloned as previously described[35]. The LRRK1 sequence was codon optimized for expression in *Spodoptera frugiperda* (Sf9) cells and synthesized by Epoch Life Science. The DNA coding for WT LRRK1 residues 631–2015 (LRRK1[RCKW]) was cloned through Gibson assembly into the pKL baculoviral expression vector (RRID: Addgene_110741), with an N-terminal His$_6$-Z-tag and TEV protease cleavage site. LRRK2 mutants were cloned using QuikChange site-directed mutagenesis (Agilent), or Q5 site-directed mutagenesis (New England Biolabs), following the manufacturer's instructions. As previously described for LRRK2[RCKW] (ref. [35]), the LRRK1[RCKW] plasmid was used for the generation of recombinant baculoviruses according to bac-to-bac expression system protocols (Invitrogen).

For mammalian expression, GFP-LRRK2 was cloned into the pDEST53 vector (RRID: Addgene_25044) as previously described[35]. LRRK2 mutants were cloned using QuikChange site-directed mutagenesis (Agilent) using standard protocols, except for liquid cultures of *Escherichia coli*, which were grown at 30 °C. EGFP-Rab10 (ref. [66]) was obtained from Addgene (RRID: Addgene_49472), and pET17b-Kif5b(1-560)-GFP-His[67] was obtained from Addgene (RRID: Addgene_15219).

### LRRK2[RCKW] and LRRK1[RCKW] expression and purification

N-terminally His$_6$-Z-tagged LRRK2[RCKW] was expressed in Sf9 insect cells (Thermo Fisher Scientific cat. no. 11496015) and purified as previously described[35]. Protocols are also available at https://doi.org/10.17504/protocols.io.rm7vzyyrrlx1/v1 and https://doi.org/10.17504/protocols.io.81wgb6693lpk/v1. Briefly, ~1 L insect cells was infected with baculovirus and grown at 27 °C for 3 days. Pelleted Sf9 cells were resuspended in lysis buffer (50 mM HEPES pH 7.4, 500 mM NaCl, 20 mM imidazole, 0.5 mM TCEP, 5% glycerol, 5 mM MgCl$_2$, 20 µM GDP, 0.5 mM Pefabloc, and protease inhibitor cocktail tablets) and lysed by Dounce homogenization. Clarified lysate was incubated with Ni-NTA agarose beads (Qiagen), extensively washed with lysis buffer, and eluted in buffer containing 300 mM imidazole. Protein eluate was diluted twofold in buffer containing no NaCl, loaded onto an SP Sepharose column, and eluted with a 250 mM to 2.5 M NaCl gradient. Protein was cleaved by TEV protease overnight. Cleaved protein was isolated by running over a second Ni-NTA column. Protein was concentrated and run on an S200 gel filtration column equilibrated in storage buffer (20 mM HEPES pH 7.4, 700 mM NaCl, 0.5 mM TCEP, 5% glycerol, 2.5 mM MgCl$_2$, 20 µM GDP). Protein was concentration to a final concentration of ~20–30 µM, as estimated by absorbance at 280 nm using an extinction coefficient of 140,150 M$^{-1}$ cm$^{-1}$.

### Purification of molecular motors

Human KIF5B$^{1-560}$(K560)-GFP was purified from *E. coli* using an adapted protocol that has been previously described[68]. (Our current protocol can also be found at https://doi.org/10.17504/protocols.io.bp2l61xrdvqe/v1.) All protein purification steps were performed at 4 °C unless otherwise noted. pET17b-Kif5b(1-560)-GFP-His was transformed into BL-21[DE3]RIPL cells (Agilent cat. no. 230280) until an optical density at 600 nm of 0.6–0.8, and expression was induced with 0.5 mM isopropyl-β-ᴅ-thiogalactoside (IPTG) for 16 hours at 18 °C. Frozen pellets from 7.5 L of culture were resuspended in 120 ml lysis buffer (50 mM Tris, 300 mM NaCl, 5 mM MgCl$_2$, 0.2 M sucrose, 1 mM dithiothreitol (DTT), 0.1 mM Mg-ATP, and 0.5 mM Pefabloc, pH 7.5) supplemented with one cOmplete EDTA-free protease inhibitor cocktail tablet (Roche) per 50 ml and 1 mg/ml lysozyme. The resuspension was incubated on ice for 30 minutes and lysed by sonication. Sonicate was supplied with 0.5 mM PMSF and clarified by centrifugation at 40,000 rcf (118,272$g$) for 60 minutes in a Type 70 Ti rotor (Beckman). The clarified supernatant was incubated with 15 ml Ni-NTA agarose (Qiagen) and rotated in a nutator for 1 hour. The mixture was washed with 100 ml

wash buffer (50 mM Tris, 300 mM NaCl, 5 mM MgCl$_2$, 0.2 M sucrose, 10 mM imidazole, 1 mM dithiothreitol (DTT), 0.1 mM Mg-ATP, and 0.5 mM Pefabloc, pH 7.5) by gravity flow. Beads were resuspended in elution buffer (50 mM Tris, 300 mM NaCl, 5 mM MgCl$_2$, 0.2 M sucrose, 250 mM imidazole, 0.1 mM Mg-ATP, and 5 mM βME, pH 8.0), incubated for 5 minutes, and eluted stepwise in 0.5-mL increments. Peak fractions were combined and buffer exchanged on a PD-10 desalting column (GE Healthcare) equilibrated in storage buffer (80 mM PIPES, 2 mM MgCl$_2$, 1 mM EGTA, 0.2 M sucrose, 1 mM DTT, and 0.1 mM Mg-ATP, pH 7.0). Peak fractions of motor solution were either flash-frozen at −80 °C until further use or immediately subjected to microtubule-bind-and-release purification. A total of 1 ml motor solution was incubated with 1 mM AMP-PNP and 20 µM taxol on ice in the dark for 5 minutes and subsequently warmed to room temperature. For microtubule bind and release, polymerized bovine brain tubulin was centrifuged through a glycerol cushion (80 mM PIPES, 2 mM MgCl$_2$, 1 mM EGTA, and 60% glycerol (vol/vol) with 20 µM taxol and 1 mM DTT) and resuspended as previously described[69], and was incubated with the motor solution in the dark for 15 minutes at room temperature. The motor–microtubule mixture was laid on top of a glycerol cushion and centrifuged in a TLA120.2 rotor at 80,000 r.p.m. (278,088$g$) for 12 minutes at room temperature. Final pellet (kinesin-bound microtubules) was washed with BRB80 (80 mM PIPES, 2 mM MgCl$_2$, and 1 mM EGTA, pH 7.0) and incubated in 100 µL of release buffer (80 mM PIPES, 2 mM MgCl$_2$, 1 mM EGTA, and 300 mM KCl, pH 7 with 7.5 mM Mg-ATP) for 5 minutes at room temperature. The kinesin release solution was spun at 72,000 r.p.m. (225,252$g$) in TLA100 for 7 minutes at room temperature. The supernatant containing released kinesin was supplemented with 660 mM sucrose and flash-frozen. A typical kinesin prep in the lab yielded 0.5 to 1.5 µM K560-GFP dimer.

### Rab8a expression and purification

Rab8a was expressed and purified as previously described[35]. The protocol is also available at https://doi.org/10.17504/protocols.io.6qpvr63mzvmk/v1. Briefly, N-terminally His$_6$-ZZ tagged Rab8a was expressed in BL21(DE3) *E. coli* cells (Agilent cat. no. 200131) by addition of 0.5 mM IPTG for 18 hours at 18 °C. Cells were pelleted, resuspended in lysis buffer (50 mM HEPES pH 7.4, 200 mM NaCl, 2 mM DTT, 10% glycerol, 5 mM MgCl$_2$, 0.5 mM Pefabloc, and protease inhibitor cocktail tablets), and lysed by sonication on ice. Clarified lysate was incubated with Ni-NTA agarose (Qiagen). Protein was washed with wash buffer (50 mM HEPES pH 7.4, 150 mM NaCl, 2 mM DTT, 10% glycerol, 5 mM MgCl$_2$) and eluted in buffer containing 300 mM imidazole. Eluate was incubated with IgG Sepharose 6 fast flow beads. Following further washing, Rab8a was cleaved off IgG sepharose beads by incubation with TEV protease at 4 °C overnight. Cleaved Rab8a was isolated by incubation with Ni-NTA agarose beads, followed by washing with buffer containing 25 mM imidazole. Purified Rab8a was run on an S200 gel filtration column equilibrated in S200 buffer (50 mM HEPES pH 7.4, 200 mM NaCl, 2 mM DTT, 1% glycerol, 5 mM MgCl$_2$). Protein was then concentrated and exchanged into 10% glycerol for storage.

### Cryo-electron microscopy: sample preparation and imaging of filaments

LRRK2[RCKW] filaments were prepared as previously described[35], with the exception that 10% glycerol was used instead of 10% DMSO in all the samples, except for the one that led to the initial data set ('19dec14f'), as glycerol promotes the formation of 11- and 12-protofilament microtubules. For '+MLi-2' samples, we added MLi-2 to LRRK2[RCKW] to a final concentration of 5 µM after incubation with tubulin. The updated protocol is also available at protocols.io (https://www.protocols.io/view/reconstituting-lrrk2rckw-on-microtubules-for-cryo-3byl4kjb8vo5/v1).

Cryo-EM data were collected on a Talos Arctica (FEI) operated at 200 kV, equipped with a K2 Summit direct electron detector (Gatan). Automated data collection was performed using Leginon[70] (version 3.4,

https://emg.nysbc.org/redmine/projects/leginon, RRID: SCR_016731) with a custom-made plug-in to automate the targeting to areas of the sample that contained LRRK2[RCKW] filaments. The only exception was the first data set ('19dec14f'), which was collected using Leginon's regular raster target finder. The '19dec14f' data set was subsequently used for training the machine-learning component of the custom-made plug-in used for all other datasets. The code for the plug-in is available at https://github.com/matyszm/filfinder (https://doi.org/10.5281/zenodo.5854954).

The 'Apo' reconstruction was obtained using two datasets: 836 micrographs from '19dec14f' and 1,010 micrographs from '20aug12b.' The 'MLi-2' reconstruction was also obtained from two datasets: 926 micrographs from '20sep10b' and 1,430 micrographs from '20sep30c.' Final micrograph counts include only micrographs with at least one usable LRRK2[RCKW] filament. The dose per data set varied between 5 and 5.5 electrons $Å^{-2} s^{-1}$. To accommodate for that range, we varied the exposure time between 10 and 11 seconds, with 200-ms frames, for a total number of frames between 50 and 55, and a total dose of 55 electrons $Å^{-2}$. The images were collected at the nominal magnification of ×36,000, resulting in an object pixel size of 1.16 Å. The defocus was set to −1.5 μm, with a final range of defoci from −0.5 to −2.5 μm owing to the nature of the lacey carbon grids and the collection strategy. All datasets are available on EMPIAR (see Table 1 for accession codes).

## Cryo-electron microscopy: reconstruction of LRRK2[RCKW] bound to a microtubule

Movie frames were aligned using UCSF MotionCor2 (ref. [71]) (version 1.4.5, https://emcore.ucsf.edu/ucsf-software, RRID: SCR_016499) with the dose-weighting option on. CTF estimation was done with CTFFIND4 (ref. [72]) (version 4.1.14, http://grigorefflab.umassmed.edu/ctffind4, RRID: SCR_016732) using the non-dose-weighted aligned micrographs. All micrographs containing filaments were kept regardless of the CTF estimated resolution. Data processing up to the symmetry expansion step is detailed in the protocol available at protocols.io (https://doi.org/10.17504/protocols.io.bwwnpfde). In brief, manual selected filaments from a subset of micrographs were 2D classified (Relion 3.1, https://github.com/3dem/relion, RRID: SCR_016274)[73], with the best classes then acting as a reference for automated filament picking (Relion 3.1). The separation distance of the particles was set to 30 Å, which ensures each particle contains one new LRRK2[RCKW] dimer per strand. These particles were then filtered first by classification on the basis of the presence of a microtubule, then followed up by another 2D classification focusing on the presence of ordered LRRK2[RCKW] filaments if MLi-2 was present, or a blurred, disordered layer if working with apo filaments. The selected particles were then 3D classified into six classes (Relion 3.1), each corresponding to a specifically sized microtubule (from 11 to 16 protofilaments). This step is inspired by MiRP[74] and used their provided reference scaled to the appropriate pixel size (https://github.com/moores-lab/MiRP, commit at time of download: 3e3b699). Filaments with MLi-2 tend to favor 11-protofilament microtubules, whereas the apo filaments favored larger sizes. We kept all the 11-protofilament microtubules for the MLi-2 dataset and all the 12-protofilament microtubules for the apo dataset.

In order to more accurately reconstruct the LRRK2[RCKW] filaments, the microtubule had to be digitally subtracted from the particles. To accomplish this, we refined the structure of the microtubule for each dataset (Relion 3.1) and subtracted it from the particles (Relion 3.1, using legacy subtraction mode). This allowed us to 2D classify (Relion 3.1) focusing on LRRK2[RCKW] filaments. Particles falling into ordered 2D classes were further 3D classified (Relion 3.1). The initial reference for each subgroup (with or without MLi-2) was always a featureless cylinder and was initialized with the helical symmetry reported for microtubule-associated LRRK2 filaments in cells[34]. Subsequent rounds used the output as the reference and were allowed to refine the symmetry, often showing multiple classes. Once the symmetry was found,

## Table 1 | Cryo-EM data collection, refinement, and validation statistics

| | LRRK2[RCKW]+MT +MLi-2 (Helical) | LRRK2[RCKW]+MT +MLi-2 (Tetramer only) |
|---|---|---|
| | EMDB-25649 | EMDB-25664 |
| | EMPIAR-10925 | EMPIAR-10925 |
| **Data collection and processing** | | |
| Magnification | 36,000 | 36,000 |
| Voltage (kV) | 200 | 200 |
| Electron exposure (e⁻/Å²) | 55 | 55 |
| Defocus range (μm) | 0.5–2.5 | 0.5–2.5 |
| Pixel size (Å) | 1.16 | 1.16 |
| Symmetry imposed | +32.5° rot, 33.3Å rise | $C_1$ |
| Initial particle images (no.) | 354,271 | 206,649 (symmetry expanded) |
| Final particle images (no.) | 14,350 | 133,246 |
| Map resolution (Å) | 18 | 5.9 |
| FSC threshold 0.143 | | |
| Map resolution range (Å) | N/A | 3.5–9 |
| **Refinement** | | |
| Initial model used (PDB code) | N/A | N/A |
| Model resolution (Å) | N/A | N/A |
| FSC threshold | | |
| Model resolution range (Å) | N/A | N/A |
| Map sharpening B factor (Å²) | N/A | −339 |
| Model composition | N/A | N/A |
| Non-hydrogen atoms | | |
| Protein residues | | |
| Ligands | | |
| B factors (Å²) | N/A | N/A |
| Protein | | |
| Ligand | | |
| R.m.s. deviations | N/A | N/A |
| Bond lengths (Å) | | |
| Bond angles (°) | | |
| Validation | N/A | N/A |
| MolProbity score | | |
| Clashscore | | |
| Poor rotamers (%) | | |
| Ramachandran plot | N/A | N/A |
| Favored (%) | | |
| Allowed (%) | | |
| Disallowed (%) | | |

| | LRRK2[RCKW]+MT + MLi-2 (tetramer+MT) | LRRK2[RCKW]+MT + MLi-2 (microtubule only) |
|---|---|---|
| | EMDB-25658 | EMDB-25908 |
| | EMPIAR-10925 | EMPIAR-10925 |
| **Data collection and processing** | | |
| Magnification | ×36,000 | ×36,000 |
| Voltage (kV) | 200 | 200 |

## Table 1 (continued) | Cryo-EM data collection, refinement, and validation statistics

| | LRRK2^RCKW+MT + MLi-2 (tetramer+MT) | LRRK2^RCKW+MT + MLi-2 (microtubule only) |
|---|---|---|
| | EMDB-25658 | EMDB-25908 |
| | EMPIAR-10925 | EMPIAR-10925 |
| Electron exposure (e⁻/Å²) | 55 | 55 |
| Defocus range (µm) | 0.5–2.5 | 0.5–2.5 |
| Pixel size (Å) | 1.16 | 1.16 |
| Symmetry imposed | $C_1$ | $C_1$ |
| Initial particle images (no.) | 206,649 (symmetry expanded) | 206,649 (symmetry expanded) |
| Final particle images (no.) | 133,246 | 133,246 |
| Map resolution (Å) | 6.6 | 5.4 |
| FSC threshold 0.143 | | |
| Map resolution range (Å) | 3.7–9.5 | 2.6–9.0 |
| **Refinement** | | |
| Initial model used (PDB code) | N/A | N/A |
| Model resolution (Å) | N/A | N/A |
| FSC threshold | | |
| Model resolution range (Å) | N/A | N/A |
| Map sharpening $B$ factor (Å²) | −326 | −235 |
| Model composition | N/A | N/A |
| Non-hydrogen atoms | | |
| Protein residues | | |
| Ligands | | |
| $B$ factors (Å²) | N/A | N/A |
| Protein | | |
| Ligand | | |
| R.m.s. deviations | N/A | N/A |
| Bond lengths (Å) | | |
| Bond angles (°) | | |
| Validation | N/A | N/A |
| MolProbity score | | |
| Clashscore | | |
| Poor rotamers (%) | | |
| Ramachandran plot | N/A | N/A |
| Favored (%) | | |
| Allowed (%) | | |
| Disallowed (%) | | |

| | LRRK2^RCKW+MT +MLi-2 (minus end) | LRRK2^RCKW+MT +MLi-2 (plus end) |
|---|---|---|
| | EMDB-25674 | EMDB-25672 |
| | EMPIAR-10924 | EMPIAR-10921 |
| | PDB-7THY | PDB-7THZ |
| **Data collection and processing** | | |
| Magnification | ×36,000 | ×36,000 |
| Voltage (kV) | 200 | 200 |
| Electron exposure (e⁻/Å²) | 55 | 55 |
| Defocus range (µm) | 0.5–2.5 | 1.2–1.8 |
| Pixel size (Å) | 1.16 | 1.16 |

| | LRRK2^RCKW+MT +MLi-2 (minus end) | LRRK2^RCKW+MT +MLi-2 (plus end) |
|---|---|---|
| | EMDB-25674 | EMDB-25672 |
| | EMPIAR-10924 | EMPIAR-10921 |
| | PDB-7THY | PDB-7THZ |
| Symmetry imposed | $C_1$ | $C_1$ |
| Initial particle images (no.) | 206,649 (symmetry expanded) | 206,649 (symmetry expanded) |
| Final particle images (no.) | 99,854 | 99,854 |
| Map resolution (Å) | 5.2 | 5.0 |
| FSC threshold 0.143 | | |
| Map resolution range (Å) | 2.6–9.0 | 2.9–7.0 |
| **Refinement** | | |
| Initial model used (PDB code) | Q5S007 (AlphaFold) | Q5S007 (AlphaFold) |
| Model resolution (Å) | 5.4 (average) | 5.3 (average) |
| Resolution method | Q-score | Q-score |
| Model resolution range (Å) | 3.0–8.6 | 3.5–8.0 |
| Map sharpening $B$ factor (Å²) | −200 | −200 |
| Model composition | N/A | N/A |
| Non-hydrogen atoms | | |
| Protein residues | | |
| Ligands | | |
| $B$ factors (Å²) | N/A | N/A |
| Protein | | |
| Ligand | | |
| R.m.s. deviations | N/A | N/A |
| Bond lengths (Å) | | |
| Bond angles (°) | | |
| Validation | N/A | N/A |
| MolProbity score | | |
| Clashscore | | |
| Poor rotamers (%) | | |
| Ramachandran plot | N/A | N/A |
| Favored (%) | | |
| Allowed (%) | | |
| Disallowed (%) | | |

| | LRRK2^RCKW+MT +MLi-2 (focused on kinase) |
|---|---|
| | EMDB-25897 |
| | EMPIAR-10925 |
| **Data collection and processing** | |
| Magnification | 36,000 |
| Voltage (kV) | 200 |
| Electron exposure (e⁻/Å²) | 55 |
| Defocus range (µm) | 0.5–2.5 |
| Pixel size (Å) | 1.16 |
| Symmetry imposed | $C_1$ |
| Initial particle images (no.) | 206,649 (symmetry expanded) |
| Final particle images (no.) | 133,246 |
| Map resolution (Å) | 4.5 |
| FSC threshold 0.143 | |

## Table 1 (continued) | Cryo-EM data collection, refinement, and validation statistics

| | LRRK2$^{RCKW}$+MT +MLi-2 (focused on kinase) |
|---|---|
| | EMDB-25897 |
| | EMPIAR-10925 |
| Map resolution range (Å) | 3.0–8.0 |
| **Refinement** | |
| Initial model used (PDB code) | N/A |
| Model resolution (Å) | N/A |
| FSC threshold | |
| Model resolution range (Å) | N/A |
| Map sharpening $B$ factor (Å$^2$) | –146 |
| Model composition | N/A |
| Non-hydrogen atoms | |
| Protein residues | |
| Ligands | |
| $B$ factors (Å$^2$) | N/A |
| Protein | |
| Ligand | |
| R.m.s. deviations | N/A |
| Bond lengths (Å) | |
| Bond angles (°) | |
| Validation | N/A |
| MolProbity score | |
| Clashscore | |
| Poor rotamers (%) | |
| Ramachandran plot | N/A |
| Favored (%) | |
| Allowed (%) | |
| Disallowed (%) | |

| | LRRK2$^{RCKW}$+MT +MLi-2 (minus end) | LRRK2$^{RCKW}$+MT +MLi-2 (plus end) |
|---|---|---|
| | EMDB-25674 | EMDB-25672 |
| | EMPIAR-10924 | EMPIAR-10921 |
| | PDB-7THY | PDB-7THZ |
| **Data collection and processing** | | |
| Magnification | 36,000 | 36,000 |
| Voltage (kV) | 200 | 200 |
| Electron exposure (e$^-$/Å$^2$) | 55 | 55 |
| Defocus range (μm) | 0.5–2.5 | 1.2–1.8 |
| Pixel size (Å) | 1.16 | 1.16 |
| Symmetry imposed | $C_1$ | $C_1$ |
| Initial particle images (no.) | 206,649 (symmetry expanded) | 206,649 (symmetry expanded) |
| Final particle images (no.) | 99,854 | 99,854 |
| Map resolution (Å) | 5.2 | 5.0 |
| FSC threshold 0.143 | | |
| Map resolution range (Å) | 2.6–9.0 | 2.9–7.0 |
| **Refinement** | | |
| Initial model used (PDB code) | Q5S007 (AlphaFold) | Q5S007 (AlphaFold) |

| | LRRK2$^{RCKW}$+MT +MLi-2 (minus end) | LRRK2$^{RCKW}$+MT +MLi-2 (plus end) |
|---|---|---|
| | EMDB-25674 | EMDB-25672 |
| | EMPIAR-10924 | EMPIAR-10921 |
| | PDB-7THY | PDB-7THZ |
| Model resolution (Å) | 5.4 (average) | 5.3 (average) |
| Resolution method | Q-score | Q-score |
| Model resolution range (Å) | 3.0-8.6 | 3.5-8.0 |
| Map sharpening $B$ factor (Å$^2$) | –200 | –200 |
| Model composition | | |
| Non-hydrogen atoms | 1012 | 1012 |
| Protein residues | 194 | 194 |
| Ligands | 1 | 1 |
| $B$ factors (Å$^2$) | | |
| Protein | –219 (average) | –216 (average) |
| Ligand | N/A | N/A |
| R.m.s. deviations | | |
| Bond lengths (Å) | 0.019 (average) | 0.020 (average) |
| Bond angles (°) | 1.907 (average) | 2.023 (average) |
| Validation | | |
| MolProbity score | 1.49 (average) | 1.54 (average) |
| Clashscore | 3.52 (average) | 3.96 (average) |
| Poor rotamers (%) | 0 (average) | 0 (average) |
| Ramachandran plot | | |
| Favored (%) | 94.5 (average) | 94.6 (average) |
| Allowed (%) | 4.9 (average) | 4.8 (average) |
| Disallowed (%) | 0.6 (average) | 0.6 (average) |

a local refinement was done with the original un-subtracted particles to give a LRRK2$^{RCKW}$ filament containing some of the original microtubule density. Since our LRRK2$^{RCKW}$ filaments each have three strands, we used symmetry expansion to extract an individual dimer from each strand. We centered the new particles on the subtraction mask and decreased the box size to 300 pixels while keeping the Å/px scale the same. This step was performed with the new subtraction function in Relion 3.1. This resulted in 206,649 particles for the MLi-2 dataset and 49,629 particles for the apo dataset. See Extended Data Figures 1b,c and 2 for the data-processing workflow. After symmetry expansion, the newly generated particles were exclusively processed in CryoSPARC[75] (version 3.2.0, https://cryosparc.com/, RRID: SCR_016501). The first step was always to align the particles to the centered subtraction mask in order to align the particles to each other. For the particles from the MLi-2 dataset, we were able to compare the Psi Euler angle to the original angle assigned during the microtubule-only alignment. Because the particles could be flipped during the LRRK2$^{RCKW}$ refinement, only particles showing 0° ± 20° and 180° ± 20° were kept. Particles with a ~180° flip were flipped back to align them to the microtubule. This left us with 133,246 particles for the MLi-2 dataset. This step was skipped for the apo particles owing to the lower particle count.

The MLi-2 dataset was processed in two ways, resulting in different levels of detail in either the kinase or ROC regions. The first processing strategy was designed to achieve a better kinase reconstruction. Here, we allowed the filtered particles to be freely aligned again, ignoring the microtubule orientation. This was followed by two local refinements: the first focused on a LRRK2$^{RCKW}$ tetramer, and the second on a single LRRK2$^{RCKW}$ monomer. The second strategy was designed to better resolve the contacts between the ROC domain and the microtubule.

Here, we performed only local refinements on the particles with the fixed microtubule orientation. To make sure the microtubule was properly aligned, we performed a local refinement focusing only on the microtubule, which resulted in a map with no ambiguity in the tubulin orientation. We then did a local refinement focused on a LRRK2[RCKW] dimer, followed by a 3D Variability Analysis (3DVA, in cryoSPARC)[76] focused on a single LRRK2[RCKW] monomer with the goal of being able to separate particles that have intact LRRK2[RCKW] in them. We analyzed the components generated and determined that components 1 and 2 ranged from a well formed LRRK2[RCKW] to having discontinuities or weak densities. We kept only particles with a negative value for at least one of the two components. Following that, we did another refinement for a LRRK2[RCKW] dimer using the filtered particles, and then used a smaller mask to cover either the '+' or '−' LRRK2[RCKW] (see Extended Data Fig. 1c) along with a small part of the microtubule. This resulted in maps showing the ROC domains interacting with the microtubule.

For the apo sample, only the freely aligned approach was used because the particle count and resolution were too low to filter the particles by microtubule orientation. After freely aligning the particles to the recentered subtraction mask, we performed a local refinement focused on the LRRK2[RCKW] tetramer. To help the alignment, we used a 20-Å low-passed LRRK2[RCKW] tetramer reference built by rigid-body fitting four copies of LRRK2[RCKW] into the early 9-Å reconstruction. This new reconstruction was still noisy, most likely owing to multiple conformations being present. Although Relion Class3D did not work on this dataset, we were able to use 3DVA again to help us find a component to separate apo LRRK2[RCKW] into classes. Component 1 resulted in a more detailed reconstruction at both positive and negative ends of the spectra than the starting structure. We reconstructed both sets, and both were able to reach ~7-Å resolution, the data with the positive component 1 resulted in a more continuous map and was chosen as the final map.

### Model creation and refinement: LRRK2 ROC domain interacting with the microtubule

For modeling, we used the maps in which the refinement had been focused on the interactions of the ROC domain with the microtubule, facing either towards the plus or minus end of the microtubule. For the initial model, we used LRRK2's model from the AlphaFold Protein Structure Database[47,48] (Q5S007) as it had the most complete loops available for the ROC domain (using residues 1332–1525). Since AlphaFold models lack ligands, we added GDP on the basis of the placement in previous structures[35,39]. Because the ROC domain occupies only a small portion of the map and some microtubule density is present, we added tubulin dimers (PDB code: 1TUB) to provide a restraint during refinement. Initial refinement was done using Rosetta (ver 3.13, https://www.rosettacommons.org/, RRID: SCR_015701) and additional refinement scripts[77]. Two hundred models were generated from each map. Tubulin dimers were removed from the model before further quantification. Models with the best energy score and fit to the density were manually inspected. Small modeling errors were corrected in Isolde[78] by hand and refined one more time in Rosetta using Relax with the map density loaded in as a restraint. Five models were selected for each map and converted to poly-alanine models except for residues of interest (K1358, K1359, R1384, K1385, R1501).

### Cryo-electron microscopy: sample preparation and imaging of LRRK1[RCKW]

The protocol for preparing LRRK1[RCKW] grids is available at protocols.io (https://doi.org/10.17504/protocols.io.b3rqqm5w). Briefly, the protein was spun down after thawing, and kept on ice until grid making. We used UltrAuFoil Holey Gold 1.2/1.3 300 mesh grids and plasma cleaned them in a Solarus II (Gatan) using the QuantiFoil Au preset. Immediately before freezing, LRRK1[RCKW] was added to 'LRRK2 buffer' (20 mM HEPES pH 7.4, 80 mM NaCl, 0.5 mM TCEP, 2.5 mM MgCl$_2$, 20 µM GDP) to the desired concentration (2–6 µM protein). We used a Vitrobot Mark IV (FEI) to freeze our samples.

Cryo-EM data were collected on a Talos Arctica (FEI) operated at 200 kV, equipped with a K2 Summit direct electron detector (Gatan). Automated data collection was performed using Leginon[62] (version 3.4, https://emg.nysbc.org/redmine/projects/leginon, RRID: SCR_016731). Reconstruction was done with four datasets ('19dec11a': 847 micrographs, '19dec21c':926 micrographs, '20sep11a': 904 micrographs, and '21jan18d': 952 micrographs). One of the datasets ('20sep11a') was collected at a 20° tilt. The exposure of the micrographs varied to achieve a total dose of 55 electrons Å$^{-2}$. The images were collected at a nominal magnification of ×36,000x resulting in an object pixel size of 1.16 Å. The defocus was set to −1.5 µm, which gave a range of defoci of −0.8 to −1.8 µm over all datasets. All datasets are available on EMPIAR (Table 1).

### Cryo-electron microscopy: reconstruction of LRRK1[RCKW]

Movie frames were aligned in cryoSPARC[75] (version 3.2.0, https://cryosparc.com/, RRID: SCR_016501) using the 'patch motion correction' program. CTF estimation was also done in cryoSPARC using the 'patch CTF estimation' program. Images were manually screened for any obvious defects and removed from further processing if defects were found. Particle picking was done with a mixture of a crYOLO[79] (version 1.6, https://cryolo.readthedocs.io/, RRID:SCR_018392) set previously trained for LRRK2[RCKW] (ref. [35]) and simple blob picking followed by a round of 2D classification to remove obvious contaminants. Both methods gave similar results, and both were used depending on whether the picking was done on the fly (blob picker) or later (crYOLO). The final particle count was 645,743.

At this point, 2D classification was used on the combined particles. Only classes showing an intact RCKW-like shape were kept. Ab initio reconstruction gave us two classes, with two-thirds of the particles ending in the intact class. We recovered additional intact particles from the broken class after another round of 2D classification. Combining class 1 and the good 2D classes gave us 131,821 particles, from which we were able to obtain a 5.8-Å map with some stretched features, likely owing to preferred orientation. To lower the impact of preferred orientation, we used 'Rebalance 2D' with the rebalance factor set to 0.7, making sure the smallest supergroup is at least 70% of the size of the largest. While the resolution dropped to 6.5 Å, the severity of the stretching was reduced.

Despite this improvement, the map contained discontinuous density on the edges of the mask, suggesting problems with the automatically generated mask. We remade the mask by basing it on homology models of LRRK1[RCKW] domains (ROC, COR, and Kinase; SWISS model[80]), and LRRK2's WD40 domain that we rigid-body fitted into the current best LRRK1[RCKW] density and used molmap in ChimeraX (ref. [81]) (version 1.2.5, https://www.cgl.ucsf.edu/chimerax/, RRID: SCR_015872) to create a map to serve as the mask. This map was low-passed to 15 Å, dilated by 8 px, and soft padded by another 8 px. This was then used to refine the structure one more time. This new map still contained artifacts in the ROC and COR-A region. We used 3DVA[76] (cryoSPARC version 3.2.0, https://cryosparc.com/, RRID: SCR_016501) to analyze the structure and found a component showing slight movement of these domains. We selected to focus on particles in the more 'closed' state. Refining these new particles gave us a better-defined map without artifacts at 5.8 Å resolution after using cryoSPARC's Non-Uniform Refinement.

### Single-molecule microscopy and motility assays

Single-molecule kinesin motility assays were performed as previously described[35]. Protocol is also available at https://doi.org/10.17504/protocols.io.ewov14qykvr2/v1. Imaging was performed with an inverted microscope (Ti-E Eclipse; Nikon) equipped with a ×100 1.49-NA oil immersion objective (Plano Apo; Nikon). The microscope was equipped with a LU-NV laser launch (Nikon), with 405 nm, 488 nm, 532 nm, 561 nm, and 640 nm laser lines. The excitation and emission paths were filtered using appropriate single bandpass filter cubes (Chroma). The emitted signals were detected using an electron multiplying CCD camera (Andor Technology, iXon Ultra 888). The *xy* position of the

stage was controlled by ProScan linear motor stage controller (Prior). Illumination and image acquisition were controlled by NIS Elements Advanced Research software (Nikon).

Single-molecule motility assays were performed in flow chambers assembled as previously described[82]. Biotin-PEG-functionalized coverslips (Microsurfaces) were adhered to glass slides using double-sided scotch tape. Each slide contained four flow chambers. Taxol-stabilized microtubules (approximately 15 mg ml$^{-1}$) with 10% biotin-tubulin and 10% Alexa 405-tubulin were prepared as previously described[82]. For each motility experiment, 1 mg ml$^{-1}$ streptavidin (in 30 mM HEPES, 2 mM magnesium acetate, 1 mM EGTA, 10% glycerol) was incubated in the flow chamber for 3 minutes. A 1:150 dilution of taxol-stabilized microtubules in motility assay buffer (30 mM HEPES, 50 mM potassium acetate, 2 mM magnesium acetate, 1 mM EGTA, 10% glycerol, 1 mM DTT and 20 μM Taxol, pH 7.4) was added to the flow chamber for 3 minutes to adhere polymerized microtubules to the coverslip. Flow chambers containing adhered microtubules were washed twice with LRRK2 buffer (20 mM HEPES pH 7.4, 80 mM NaCl, 0.5 mM TCEP, 5% glycerol, 2.5 mM MgCl$_2$, and 20 μM GDP). Flow chambers were then incubated for 5 minutes with either LRRK2 buffer alone or LRRK2 buffer containing the indicated concentration of WT or mutant LRRK2$^{RCKW}$. Before the addition of kinesin motors, the flow chambers were washed three times with motility assay buffer containing 1 mg ml$^{-1}$ casein. The final imaging buffer for motors contained motility assay buffer supplemented with 71.5 mM βME, 1 mM Mg-ATP, and an oxygen scavenger system, 0.4% glucose, 45 μg/ml glucose catalase (Sigma-Aldrich), and 1.15 mg/ml glucose oxidase (Sigma-Aldrich). The final concentration of kinesin in the motility chamber was 1 nM. K560-GFP was imaged every 500 ms for 2 minutes with 25% laser (488) power at 150-ms exposure time. Each sample was imaged no longer than 15 minutes. Each technical replicate consisted of movies from at least two fields of view containing between five and ten microtubules each.

### Single-molecule motility assay analysis

Kymographs were generated from motility movies using ImageJ macros as described previously[82]. See https://doi.org/10.17504/protocols.io.ewov14qykvr2/v1 for a summary. Specifically, maximum-intensity projections were generated from time-lapse sequences to define the trajectory of particles on a single microtubule. The segmented line tool was used to trace the trajectories and map them onto the original video sequence, which was subsequently re-sliced to generate a kymograph. Brightness and contrast were adjusted in ImageJ for all videos and kymographs. Motile and immotile events (>1 second) were manually traced using ImageJ and quantified for run lengths and percentage motility. Run-length measurements were calculated from motile events only. For percent motility per microtubule measurements, motile events (>1 second and >785 nm) were divided by total events per kymograph. Bright aggregates, which were less than 5% of the population, were excluded from the analysis. Data visualization and statistical analyses were performed in GraphPad Prism (version 9.2, https://www.graphpad.com/, RRID: SCR_002798) and ImageJ[83] (version 1.53, https://imagej.nih.gov/ij/, RRID: SCR_003070). All tabular data are available at https://doi.org/10.5281/zenodo.6463635.

### Microtubule sedimentation binding assay

Protocol is also available at https://doi.org/10.17504/protocols.io.36wgq73b5vk5/v1. Porcine brain tubulin was purchased from Cytoskeleton Taxol-stabilized microtubules were polymerized at a final concentration of ~2.5 mg/mL, and free tubulin was removed by ultracentrifugation at 108,628 g for 15 minutes at 37 °C through a 64% glycerol cushion. The resulting microtubule pellet was resuspended in LRRK2 binding buffer (20 mM HEPES pH 7.4, 110 mM NaCl, 0.5 mM TCEP, 5% glycerol, 2.5 mM MgCl$_2$, 20 μM GDP, and 20 μM taxol). Tubulin concentration was determined by comparison of the polymerized microtubule stock to actin standards on SDS–PAGE.

For a typical LRRK$^{RCKW}$ microtubule co-sedimentation assay, 200 nM LRRK$^{RCKW}$ was incubated at room temperature for 10 minutes with various concentrations of microtubules in buffer containing 20 mM Hepes pH 7.4, 110 mM NaCl, 0.5 mM MgCl$_2$, 0.5 mM TCEP, 5% glycerol, 20 μM GDP, and 20 μM taxol. Microtubules were then pelleted by ultracentrifugation (15 minutes, 108,628 g, 25 degrees). To quantify the depletion of LRRK2$^{RCKW}$, samples of the supernatant were taken and boiled for 10 minutes in SDS buffer. Samples were run on 4–12% polyacrylamide gels (NuPage, Invitrogen) and stained with SYPRO-Red Protein Gel Stain (Thermo Fisher) for protein detection. Binding curves were fit in GraphPad Prism (version 9.2, https://www.graphpad.com/, RRID: SCR_002798) with a nonlinear regression hyperbolic curve. All tabular data are available at https://doi.org/10.5281/zenodo.6463635.

### Microtubule cleavage by subtilisin

Taxol-stabilized microtubules with 10% biotin-tubulin and 10% Alexa 488-tubulin were prepared as described in https://doi.org/10.17504/protocols.io.bp2l6bdedgqe/v1. Subtilisin (Sigma-Aldrich P580), or an equivalent volume buffer, was added to 0.7 mg/mL and incubated for 10 minutes at 37 °C. The cleavage reaction was quenched by addition of 2.8 mM PMSF. To remove free tubulin and protease, the reaction mixture was centrifuged at 108,628 g for 15 minutes at 37 °C through a 60% glycerol cushion. The microtubule pellet was resuspended in ×1 BRB80 (80 mM PIPES, 2 mM MgCl$_2$, and 1 mM EGTA, pH 7.0) with 1 mM DTT and 20 μM taxol. Cleavage was confirmed by SDS–PAGE electrophoresis (Extended Data Fig. 4o).

### TMR labeling

BODIPY TMR-X NHS Ester (Thermo Fisher) was used to fluorescently label LRRK2$^{RCKW}$ and LRRK1$^{RCKW}$. For a typical 40 μL labeling reaction, dye was added at a ratio of 1:1 to ~20 μM LRRK2$^{RCKW}$, followed by incubation at room temperature for 1 hour. Excess dye was removed by two consecutive buffer exchanges through Micro Bio-Spin P-6 desalting columns (Bio-Rad). Protein concentration and labeling efficiency were estimated using a NanoDrop Microvolume Spectrophotometer. The protocol is also available at https://doi.org/10.17504/protocols.io.ewov1nq5ogr2/v1.

### Widefield fluorescence microtubule binding assay

Imaging was performed with an inverted microscope (Nikon, Ti-E Eclipse) equipped as described above (single-molecule microscopy and motility assays). See https://doi.org/10.17504/protocols.io.kxygxz7bdv8j/v1 for an online version of this protocol.

LRRK2$^{RCKW}$ microtubule-binding experiments were performed in flow chambers made as described above (single-molecule microscopy and motility assays). LRRK$^{RCKW}$ was labeled with TMR (TMR labeling, above); taxol-stabilized microtubules were polymerized from a mixture of unmodified, biotinylated, and Alexa Fluor 488-labeled bovine tubulin, as previously described (REF). To attach microtubules to the coverslip, flow chambers were incubated with 0.5 mg/mL streptavidin for 3 minutes, washed twice in buffer (30 mM Hepes pH 7.4, 50 mM potasium acetate, 2 mM magnesium acetate, 1 mM EGTA, 10% glycerol, 1 mM DTT, and 0.2 mM taxol), and then incubated with microtubules for 3 minutes. Microtubules were washed twice in buffer (20 mM Hepes pH 7.4, 80 mM NaCl, 0.5 mM MgCl$_2$, 0.5 mM TCEP, 5% glycerol, 20 μM GDP), and then incubated with varied concentrations of LRRK2$^{RCKW}$ (6.25 nM–50 nM) for 5 minutes. Multiple fields of view were imaged along the flow chamber with the objective in widefield illumination, with successive excitation at 488 nm (15% laser power, 100 ms exposure) and 561 nm (25% laser power, 100 ms exposure).

Image analysis was performed with ImageJ[83] (version 1.53, https://imagej.nih.gov/ij/, RRID: SCR_003070). Average TMR-LRRK2$^{RCKW}$ fluorescence intensity per microtubule was calculated from a 1-pixel-wide line drawn along the long axis of the microtubule; overall average background fluorescence intensity was subtracted. These

background-subtracted intensities were averaged over all microtubules per field of view, normalized by microtubule length, to yield a single data point. Eight fields of view at each concentration of LRRK2[RKCW] were then averaged. All tabular data are available at https://doi.org/10.5281/zenodo.6463635.

## In vitro Rab8a phosphorylation

LRRK kinase assays were performed as previously described[35] with LRRK[RCKW] and Rab8a purified as described above. For a typical kinase reaction, 38 nM LRRK[RCKW] was incubated with 3.8 µM Rab8a for 30 minutes at 30° in buffer containing 50 mM Hepes pH 7.4, 80 mM NaCl, 10 mM MgCl$_2$, 1 mM ATP, 200 uM GDP, 0.5 mM TCEP. Phosphorylation of Rab8a at residue T72 by LRRK[RCKW] was monitored by western blot using a commercially available antibody (Abcam cat. no. ab230260, RRID: AB_2814988) as previously described[35,84].

## Immunofluorescence, confocal microscopy, and image analysis

LRRK2 filament assays were performed as previously described[35]. Briefly, cells were plated on fibronectin-coated glass coverslips and grown for 24 hours before transfection with PEI. Cells were transfected with 500 ng of indicated GFP-LRRK2 plasmids. After 24–48 hours, cells were incubated at 37 °C with DMSO or MLi-2 (500 nM) for 2 hours. Stocks of the kinase inhibitor MLi-2 (10 mM; Tocris) were stored in DMSO at −20 °C.

Cells were rinsed briefly with ice-cold 1× PBS on ice, then fixed with ice-cold 4% PFA, 90% methanol, and 5 MM sodium bicarbonate for 10 minutes at −20 °C. Coverslips were subsequently washed three times with ice-cold PBS and then incubated with blocking buffer (1% BSA, 5% normal goat serum, 0.3% Triton X-100 in 1× PBS) for 1 hour at room temperature. Primary antibodies were diluted in antibody dilution buffer (1% BSA, 0.1% Triton X-100 in 1× PBS) and incubated at 4 °C overnight. The following day, coverslips were washed three times with 1× PBS and incubated with secondary antibodies diluted in antibody dilution buffer for 1 hour at room temperature. After secondary incubation, coverslips were washed three times with 1× PBS. Cells were briefly rinsed in ddH$_2$O and mounted on glass slides using CitiFluor AF-1 mounting medium (TedPella). Coverslips were sealed with nail polish and stored at 4 °C. Antibodies used for immunofluorescence were used at a 1:500 dilution and included: chicken anti-GFP (Aves Labs cat. no. GFP-1020, RRID: AB_1000024) and goat anti-chicken-Alexa Fluor 488 (Thermo Fisher Scientific cat. no. A-11039, RRID:AB_2534096).). DAPI was used at 1:5,000, according to the manufacturer's recommendation (Thermo Fisher Scientific cat. no. D1306, RRID: AB_2629482).

For the LRRK2 filament analysis, experimenters were blinded to conditions for both the imaging acquisition and analysis. Cells were imaged using a Yokogawa W1 confocal scanhead mounted to a Nikon Ti2 microscope with an Apo ×60 1.49-NA objective. The microscope was run with NIS Elements using the 488 nm and 405 nm lines of a six-line (405 nm, 445 nm, 488 nm, 515 nm, 561 nm, and 640 nm) LUN-F-XL laser engine and a Prime95B camera (Photometrics).

ImageJ[83] (version 1.53, https://imagej.nih.gov/ij/, RRID: SCR_003070) was used to quantify the percentage of cells with LRRK2 filaments, as previously described. Maximum-intensity projections were generated from z-stack confocal images. Using the GFP immunofluorescence signal, transfected cells were identified. Cells were scored for the presence or absence of filaments using both the z-projection and z-stack micrographs as a guide. To calculate the percentage cells with filaments, the number of cells with filaments was divided by the total number of transfected cells per technical replicate (defined as one 24-well coverslip). Per coverslip, eight fields of view were imaged containing a total of 50 and 150 cells per replicate. The quantification of all cellular experiments come from compiled data collected on at least three separate days. All statistical analyses were performed in GraphPad Prism (version 9.2, https://www.graphpad.com/, RRID:SCR_002798).

## Western blot analysis and antibodies

For western blot quantification of LRRK2 protein expression and Rab10 phosphorylation, cells were plated on 6-well dishes (200,000 cells per well) 24 hours before transfection. Cells were transfected with 500 ng of GFP-LRRK2 construct and 500 ng of GFP-Rab10 using polyethylenimine (PEI, Polysciences). After 36 hours, cells were rinsed with ice-cold 1× PBS, pH 7.4 and lysed on ice in RIPA buffer (50 nM Tris pH7.5, 150 mM NaCl, 0.2% Triton X-100, 0.1% SDS, with cOmplete protease inhibitor cocktail and PhoStop phosphatase inhibitor). Lysates were rotated for 15 minutes at 4 °C and clarified by centrifugation at maximum speed in a 4 °C microcentrifuge for 15 minutes. Supernatants were then boiled for 10 minutes in SDS buffer. Experiments were performed in duplicate or triplicate and repeated on at least 3 separate days.

Lysates were run on 4–12% polyacrylamide gels (NuPage, Invitrogen) for 50 minutes at 180 V and transferred to polyvinylidene difluoride (Immobilon-FL, EMD Millipore) for 4 hours at 200 mA constant current. Blots were rinsed briefly in MilliQ water and dried at room temperature for at least 30 minutes. Membranes were briefly reactivated with methanol and blocked for 1 hour at room temperature in 5% milk (wt/vol) in TBS. Antibodies were diluted in 1% milk in TBS with 0.1% Tween-20 (TBST). Primary antibodies used for immunoblots were as follows: mouse anti-GFP (1:2,500 dilution; Santa Cruz Biotechnology cat. no. sc-9996, RRID:AB_627695), rabbit anti-LRRK2 (1:5,000 dilution; Abcam cat. no. ab181386), rabbit anti-GAPDH (1:3,000 dilution; Cell Signaling Technology cat. no. 2118, RRID: AB_561053), recombinant rabbit anti-phospho-T72-RAB8 (1:1,000 dilution; Abcam cat. no. ab230260, Lot: GR3216587-1), and rabbit anti-phospho-T73-RAB10 (1:2,500 dilution; Abcam cat. no. ab230261, RRID: AB_2811274). Secondary antibodies (1:15000) used for western blots were IRDye goat anti-mouse 680RD (LI-COR Biosciences cat. no. 926-68070, RRID:AB_10956588) and IRDye goat anti-rabbit 800CW (LI-COR Biosciences cat. no. 926-32211, Lot: C90229-05, RRID: AB_10956166). Primary antibodies were incubated overnight at 4 °C, and secondary antibodies were incubated at room temperature for 1 hour. For quantification, blots were imaged on an Odyssey CLx controlled by Imaging Studio software (version 5.2, https://www.licor.com/bio/image-studio/, RRID: SCR_015795), and intensity of bands quantified using Image Studio Lite software (version 5.2, RRID: SCR_013715). All tabular data are available at https://doi.org/10.5281/zenodo.6463635.

## Cell line

Human 293T cells were obtained from ATCC (ATCC cat. no. CRL-3216, RRID: CVCL_0063) and maintained at 37 °C with 5% CO$_2$ in Dulbecco's Modified Eagle Medium (DMEM, Corning) supplemented with 10% fetal bovine serum (FBS, Gibco) and 1% penicillin–streptomycin (Corning). Cells were routinely tested for mycoplasma contamination and were not authenticated after purchase.

## Sequence alignment

Protein sequences of LRRK2 (Q5S007) and LRRK1 (Q38SD2) were obtained from UniProt. Sequence alignments were performed with Clustal Omega web services[85] (https://www.ebi.ac.uk/Tools/msa/clustalo/, RRID: SCR_001591) and annotated using Jalview[86] (version 2.11, http://www.jalview.org/, RRID: SCR_006459).

## Reporting summary

Further information on research design is available in the Nature Portfolio Reporting Summary linked to this article.

## Data availability

Cryo-EM maps and molecular models have been deposited in the EM Data Bank and Protein Data Bank, respectively. EMDB accession numbers are as follows: EMDB-25649: LRRK2[RCKW] + MT + MLi-2 (helical); EMDB-25658: LRRK2[RCKW] + MT + MLi-2 (tetramer + MT);

EMDB-25664: LRRK2$^{RCKW}$ + MT + MLi-2 (tetramer only); EMDB-25672: LRRK2$^{RCKW}$ + MT + MLi-2 (plus end); EMDB-25674: LRRK2$^{RCKW}$ + MT + MLi-2 (minus end); EMDB-25897: LRRK2$^{RCKW}$ + MT + MLi-2 (focused on kinase); EMDB-25908: LRRK2$^{RCKW}$ + MT + MLi-2 (microtubule only). PDB accession numbers are as follows: PDB-7THY: LRRK2$^{RCKW}$ + MT + MLi-2 (minus end); PDB-7THZ: LRRK2$^{RCKW}$ + MT + MLi-2 (plus end). Movie frames for the micrographs used in this study were deposited in EMPIAR; accession numbers are also listed in Table 1. All reagents and data will be made available upon request by the corresponding authors. Source data are provided with this paper.

## Code availability

The code for the plug-in developed to automate the targeting to areas of the sample that contained LRRK2$^{RCKW}$ filaments during data collection in Leginon is available at https://github.com/matyszm/filfinder (https://doi.org/10.5281/zenodo.5854954).

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

## Acknowledgements

This research was funded in part by Aligning Science Across Parkinson's (grant number ASAP-000519) through the Michael J. Fox Foundation for Parkinson's Research (MJFF) (A. E. L. and S. L. R.-P.). The work was also funded by the Howard Hughes Medical Institute (where S. L. R.-P. is an investigator); the Michael J. Fox Foundation (grant number 18321 to A. E. L. and S. L. R.-P.); the A.P. Giannini Foundation (postdoctoral fellowship to D. M. S.); the Molecular Biophysics Training Grant at UC San Diego (NIH Grant T32 GM008326 supported A. M. D.); and the National Institutes of Health (R35GM141825 to S. L. R.-P. and R01GM107214 to A. E. L). We also thank the UC San Diego Cryo-EM Facility, the Nikon Imaging Center at UC San Diego and Eric Griffis, and the UC San Diego Physics Computing Facility for IT support. For the purpose of open access, the authors have applied a CC BY 4.0 public copyright license to all author accepted manuscripts arising from this submission.

## Author contributions

M. M. collected and processed the LRRK2 cryo-EM data. D. M. S. and M. M. collected and processed the LRRK1 cryo-EM data. Y. X. L., D. M. S., A. M. D. and M. M. purified proteins for cryo-EM and biochemistry experiments. D. M. S. and A. M. D. performed the biochemical and single-molecule assays with the help of Y. X. L. A. M. D. performed the cellular assays. S. L. R.-P. and A. E. L. directed and supervised the research. D. M. S., M. M., A. M. D., A. E. L., and S. L. R.-P. wrote and edited the manuscript.

## Competing interests

The authors have no competing interests to declare.

## Additional information

**Extended data** is available for this paper at https://doi.org/10.1038/s41594-022-00863-y.

**Correspondence and requests for materials** should be addressed to Andres E. Leschziner or Samara L. Reck-Peterson.

**Peer review information** *Nature Structural & Molecular Biology* thanks Dario Alessi and the other, anonymous, reviewer(s) for their contribution to the peer review of this work. Editor recognition statement (if applicable to your journal): Primary Handling editor: Florian Ullrich, in collaboration with the *Nature Structural & Molecular Biology* team. Peer reviewer reports are available.

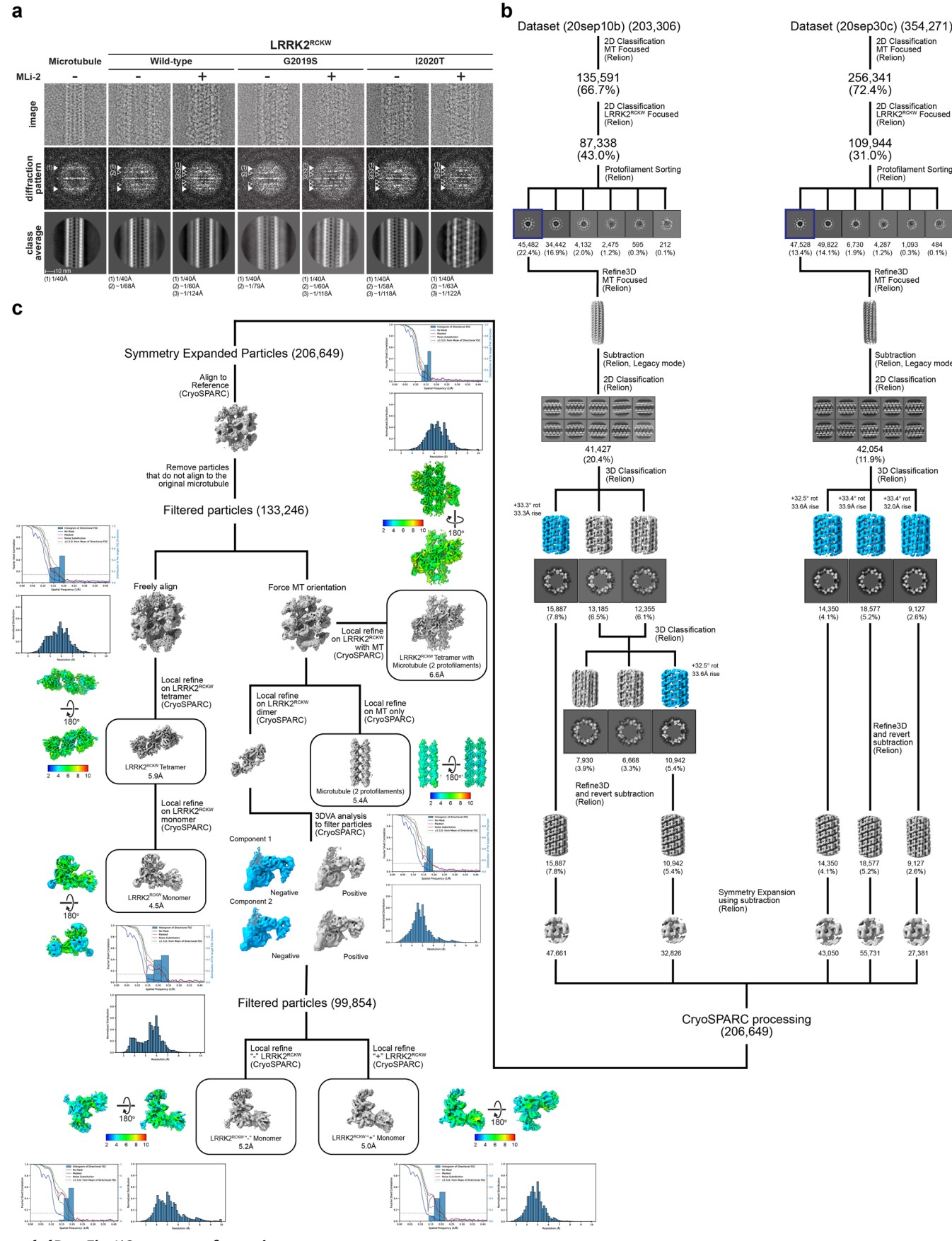

**Extended Data Fig. 1 | See next page for caption.**

**Extended Data Fig. 1 | Cryo-EM structure determination of microtubule-associated filaments of LRRK2RCKW[I2020T] in the presence of MLi-2.**
**a**, Optimization of in vitro reconstituted microtubule-associated LRRK2RCKW filaments. Top row, cryo-EM images of an individual microtubule (left) or individual microtubule-associated LRRK2RCKW filaments. Middle, Diffraction patterns calculated from the images above. Arrowheads point to layer lines arising from the microtubule (white) or from the LRRK2RCKW filaments (open), and their frequencies are indicated below the images. Bottom, 2D class averages from multiple images equivalent to those shown at the top. The type of LRRK2RCKW (WT, G2019S, or I2020T) and the presence or absence of MLi-2 during filament reconstitution are indicated at the top. **b**, Schematic of data processing pipeline used to obtain the different reconstructions of the microtubule-associated LRRK2RCKW[I2020T] filaments in the presence of MLi-2 (see Methods for details). Local resolution maps, Fourier Shell Correlation, directional FSC plots, and the distribution of voxel resolutions are shown for all reconstructions discussed in the text.

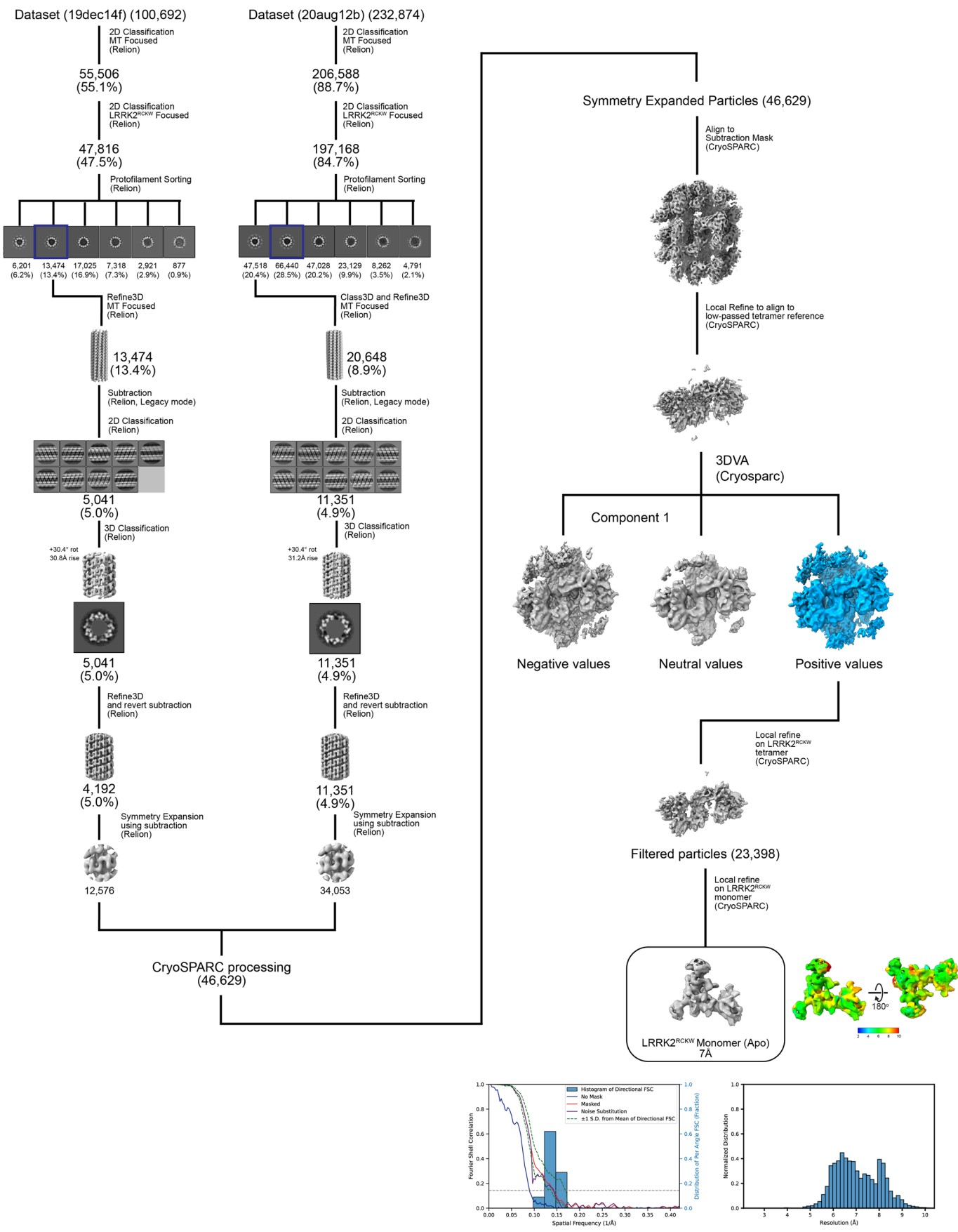

**Extended Data Fig. 2 | Cryo-EM structure determination of microtubule-associated filaments of LRRK2RCKW[I2020T] in the absence of MLi-2.** Schematic of data processing pipeline used to obtain the reconstruction of microtubule-associated LRRK2RCKW[I2020T] filaments in the absence of MLi-2 (see Methods for details). Local resolution maps, Fourier Shell Correlation, directional FSC plots, and the distribution of voxel resolutions are shown for all reconstructions discussed in the text.

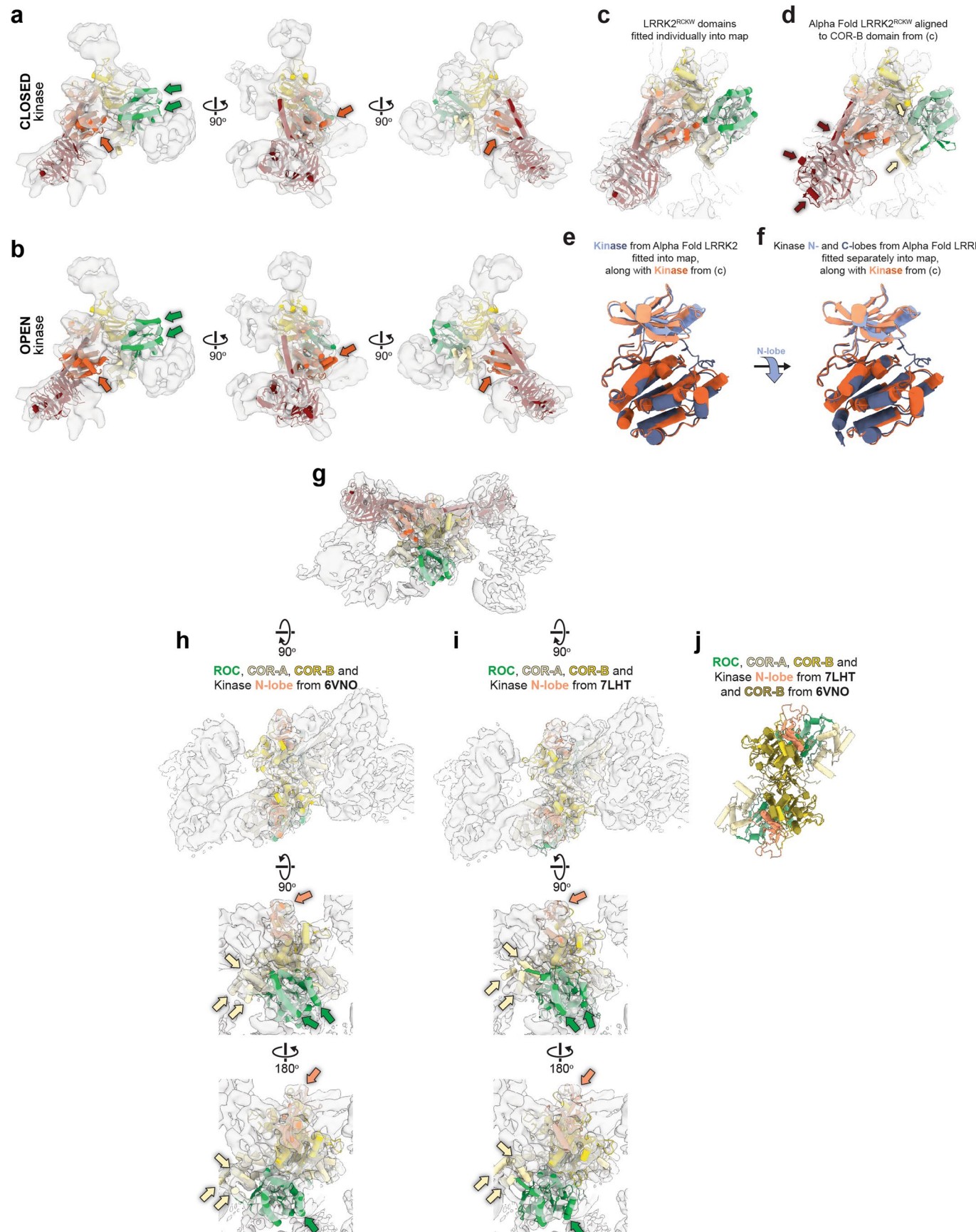

**Extended Data Fig. 3 | See next page for caption.**

**Extended Data Fig. 3 | Structural analysis of microtubule-associated filaments of LRRK2<sup>RCKW</sup>[I2020T]. a**, A model for LRRK2<sup>RCKW</sup>[I2020T] with a closed kinase, obtained by docking the individual domains into the cryo-EM map of LRRK2<sup>RCKW</sup>[I2020T] filaments obtained in the presence of MLi-2, was docked into a cryo-EM map (7 Å) of filaments obtained in the absence of the inhibitor (Extended Data Fig. 2). **b**, A model for LRRK2<sup>RCKW</sup> with an open kinase (PDB:6VNO) was docked into the same map. The colored arrows in (**a**) and (**b**) highlight structural elements in the model that protrude from the density when the kinase is in an open conformation. **c**, The LRRK2<sup>RCKW</sup> domains (ROC, COR-A, COR-B, Kinase N-lobe, Kinase C-lobe, WD40) (PDB:6VNO) were fitted individually into one of the monomers in the cryo-EM map of microtubule-bound LRRK2<sup>RCKW</sup>[I2020T] formed in the presence of MLi-2. **d**, The LRRK2<sup>RCKW</sup> portion of the AlphaFold model of LRRK2 was aligned to the COR-B domain in (**c**) and is shown here inside the same cryo-EM map. The colored arrows highlight regions where part of the model protrudes from the density. (Note: there is no arrow pointing to the loop in the ROC domain as this loop was not seen or modeled in the microtubule-bound structure). **e**, The kinase from the AlphaFold model of LRRK2 was fitted into the cryo-EM map (same as in (**d**)) and is shown here superimposed on the N- and C-lobes of the kinase as fitted in (**c**). Note that while

the C-lobes superimpose well, the N-lobe fitted individually in (**c**) is more closed than that modeled in the AlphaFold LRRK2. **f**, The N- and C-lobes of the kinase from the AlphaFold LRRK2 model were now fitted individually into the cryo-EM map (as in (**c**)) and are shown superimposed on the N- and C-lobes of LRRK2<sup>RCKW</sup> from (**a**). The blue arrow between panels (**e**) and (**f**) highlights the downward movement of the N-lobe of AlphaFold's LRRK2 when the two lobes are fitted individually into the cryo-EM map. **g**, The LRRK2<sup>RCKW</sup> domains (ROC, COR-A, COR-B, Kinase N-lobe, Kinase C-lobe, WD40) (PDB:6VNO) were fitted individually into the central dimer of the cryo-EM map of a tetramer of microtubule-bound LRRK2<sup>RCKW</sup>[I2020T] obtained in the presence of MLi-2. **h, i**, Different closeup views of the map in (**g**), showing either (**h**) the ROC, COR-A, COR-B and kinase N-lobe from the LRRK2<sup>RCKW</sup> model (PDB:6VNO), or (**i**) the corresponding portion from the structure of full-length LRRK2 (PDB:7LHT) docked as a single body into the cryo-EM map. The colored arrows highlight parts of the model that fit the cryo-EM density better when the domains are fitted in individually (**h**) rather than as a rigid body (**i**). **j**, Superposition of the model used in (**i**) and the COR-B domain from (**h**) to show that the differences among the ROC, COR-A and N-lobe of the kinase between the two models ((**h**) and (**i**)) is not due to major differences at the COR-B:COR-B interface, which is similar.

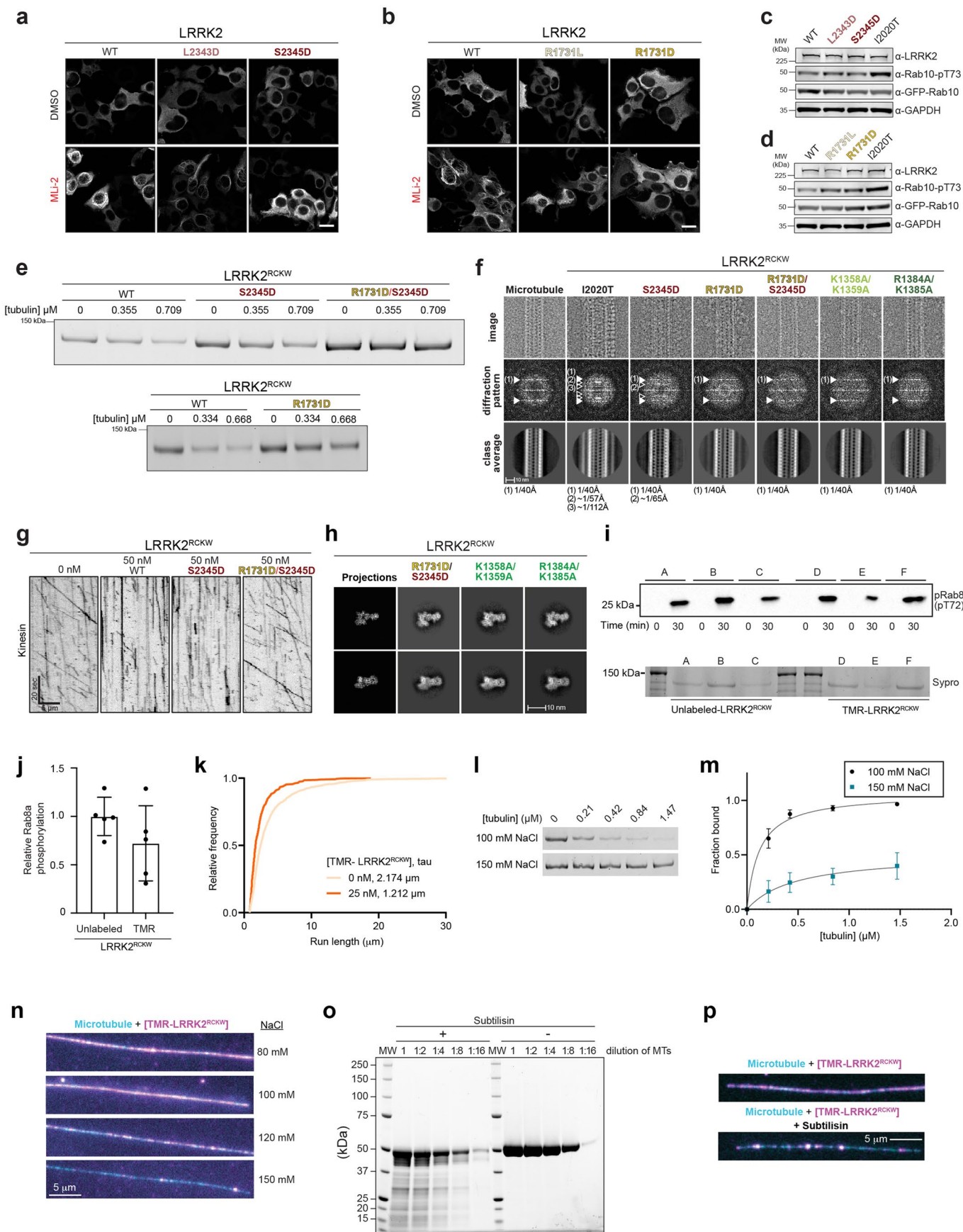

**Extended Data Fig. 4 | See next page for caption.**

**Extended Data Fig. 4 | Mechanism of LRRK2$^{RCKW}$ binding to microtubules.**
**a**, Representative images of 293 T cells expressing GFP-LRRK2 (wild-type or the WD40 mutants L23443D and S2345D) and treated with DMSO or 500 nM MLi-2 for 2 hours. Scale bar is 10 μm. **b**, Representative images of 293 T cells expressing GFP-LRRK2 (wild-type or the COR-B mutants R1731L and R1731D) and treated with DMSO or 500 nM MLi-2 for 2 hours. Scale bar is 10 μm. **c,d**, Rab10 phosphorylation in 293 T cells overexpressing WT LRRK2 or LRRK2 carrying mutations in the WD40 (**c**) or COR-B (**d**) domains. LRRK2[I2020T], which is known to increase Rab10 phosphorylation in cells, was tested as well. 293 T cells were transiently co-transfected with the indicated plasmids encoding for GFP-LRRK2 (wild type or mutant) and GFP-Rab10. Thirty-six hours post-transfection the cells were lysed, immunoblotted for phospho-Rab10 (pT73), total GFP-Rab10, and total GFP-LRRK2. Quantification of data in (**c**) and (**d**) is shown in Fig. 2d, f, respectively. **e**, A representative gel of the supernatants from the microtubule pelleting assays used to generate the data shown in Fig. 2g. **f**, Cryo-EM analysis of filament formation in vitro by LRRK2$^{RCKW}$ mutants. Top row, cryo-EM images of an individual microtubule (left) or combinations of microtubules and LRRK2$^{RCKW}$ mutants. Middle, Diffraction patterns calculated from the images above. Arrowheads point to layer lines arising from the microtubule (white) or from the LRRK2$^{RCKW}$ filaments (open), and their frequencies are indicated below the images. Bottom, 2D class averages from multiple images equivalent to those shown at the top. **g**, Example kymographs of single-molecule kinesin motility assays in the presence or absence of 50 nM LRRK2$^{RCKW}$ wild-type or indicated mutant. **h**, Comparison of 2D class averages from cryo-EM images of different LRRK2$^{RCKW}$ mutants with the corresponding 2D projection from a LRRK2$^{RCKW}$ molecular model (PDB: 6VNO). Two different views are shown for each mutant. **i**, Representative in vitro kinase reaction. Rab8a phosphorylation was measured via western blotting with a phospho-T72-specific Rab8a antibody, and total

LRRK2$^{RCKW}$ concentration was measured by Sypro Red staining. Phosphorylation reactions were terminated after 30 minutes. **j**, Quantification of data shown in (i). For each reaction, phospho-Rab8a band intensity (chemiluminescence) was divided by LRRK2$^{RCKW}$ band intensity (Sypro red); for each western blot, an average normalized value was calculated for all replicates of unlabeled LRRK2$^{RCKW}$, and all data was then normalized to this value. Data are mean +/− s.d., n = 5 replicates, p=ns, two-tailed unpaired t-test. **k**, Cumulative distribution of run lengths for kinesin in the absence or presence of 25 nM TMR-LRRK2$^{RCKW}$. The run lengths were significantly different between 0 nM and 25 nM TMR-LRRK2$^{RCKW}$ (Mann-Whitney test). Mean decay constants (tau) are shown. The effect on kinesin motility is similar to that previously shown using unlabeled LRRK2$^{RCKW}$. **l, m**, Representative microtubule pelleting assay gel for LRRK2$^{RCKW}$ in the presence of 100 mM and 150 mM sodium chloride. Co-sedimentation was measured as depletion from the supernatant. For each reaction, 200 nM LRRK2$^{RCKW}$ was mixed with a given concentration of microtubules, microtubules were pelleted by a high-speed spin, and a gel sample was taken of the supernatant. Quantification of data represented in (**l**) is shown in (**m**). Data are mean ± s.d., n = 4. The solid line represents a hyperbolic curve fit to the data. **n**, Representative images of coverslip-tethered Alexa Fluor 488-labeled MTs (cyan) bound to 100 nM TMR-LRRK2$^{RCKW}$ (magenta) in the presence of increasing concentrations of sodium chloride, used to generate the data in Fig. 3d. **o**, Subtilisin treatment of taxol-stabilized microtubules. Serial dilutions of taxol-stabilized microtubules were treated with subtilisin (left) or left untreated as a control (right). The experiment was performed twice with similar results. **p**, Representative images of untreated (top) and subtilisin-treated (bottom) Alexa Fluor 488-labeled MTs (cyan) bound to 50 nM TMR-LRRK2$^{RCKW}$ (magenta), used to generate the data shown in Fig. 3e.

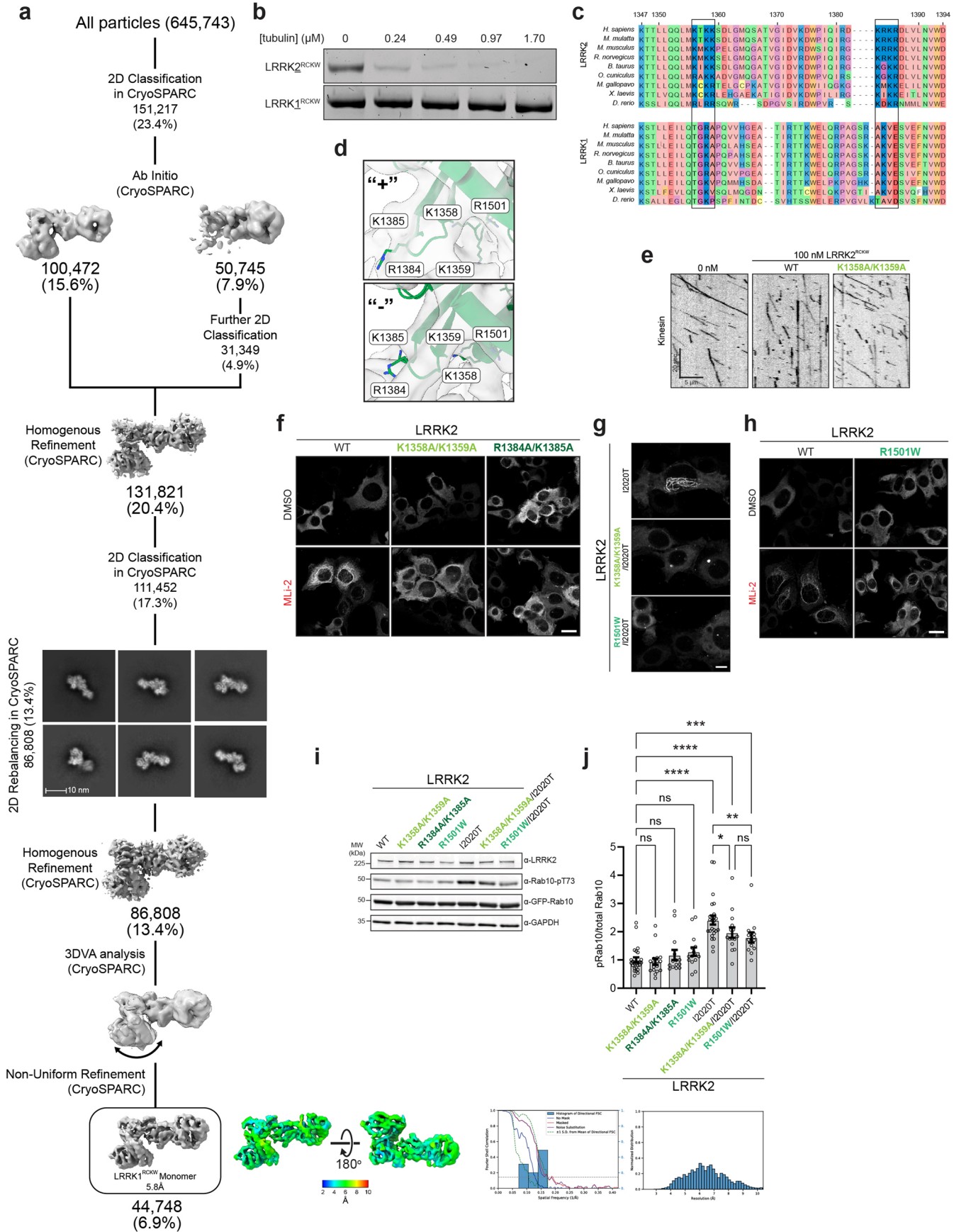

**Extended Data Fig. 5 | See next page for caption.**

**Extended Data Fig. 5 | Basic residues within the LRRK2 ROC domain are not conserved in LRRK1 and are involved in LRRK2's binding to microtubules.**
**a**, Cryo-EM structure determination of LRRK1^RCKW. Local resolution map, Fourier Shell Correlation, directional FSC plot, and the distribution of voxel resolutions are shown. **b**, Representative gel of supernatant from microtubule pelleting assay for 200 nM LRRK2^RCKW or LRRK1^RCKW with increasing tubulin concentrations. We performed 4 technical replicates. **c**, Sequence alignment of the ROC domains of LRRK2 and LRRK1 across several species made using Clustal Omega. Putative microtubule-contacting residues conserved in LRRK2 but not in LRRK1 are boxed. **d**, Close ups of the basic patches tested in this study, shown in the context of the cryo-EM maps for the '+' and '−' LRRK2^RCKW monomers in our reconstruction of the microtubule-associated filaments (Fig. 1e, f). The models shown here correspond to those in Fig. 5c. **e**, Example kymographs of kinesin motility in the presence of 100 nM LRRK2^RCKW (wild-type or K1358A/K1359A mutant). **f**, Representative images of 293 T cells expressing GFP-LRRK2 (wild type or ROC mutants) and treated with DMSO or 500 nM MLi-2 for 2 hours, corresponding to data plotted in Fig. 5g. Scale bar is 10 µm. **g**, Representative

images of 293 T cells expressing GFP-LRRK2 (I2020T, I2020T/ROC mutant, or I2020T/R1501W), corresponding to data plotted in Fig. 5h, j. Scale bar is 10 µm. **h**, Representative images of 293 T cells expressing GFP-LRRK2 (wild type or R1501W mutant) and treated with DMSO or 500 nM MLi-2 for 2 hours, corresponding to data plotted in Fig. 5i. Scale bar is 10 µm. **i, j**, Rab10 phosphorylation in 293 T cells overexpressing WT LRRK2 or LRRK2 carrying the indicated mutations in the ROC domain. LRRK2[I2020T], which is known to increase Rab10 phosphorylation in cells, was tested as well. 293 T cells were transiently transfected with the indicated plasmids encoding for GFP-LRRK2 (wild-type or mutant) and GFP-Rab10. Thirty-six hours post-transfection the cells were lysed, immunoblotted for phospho-Rab10 (pT73), total GFP-Rab10, and total GFP-LRRK2. Quantification of immunoblotting data (**i**) is shown in (**j**) as ratios of pRab10/total GFP-Rab10 normalized to the average of all wildtype values. Individual data points represent separate populations of cells obtained across at least three independent experiments. Data are mean +/− s.e.m. ****p < 0.0001, ***p = 0.0004, **p = 0.0052, ***p = 0.0344, one-way ANOVA followed by a Fisher's LSD test.

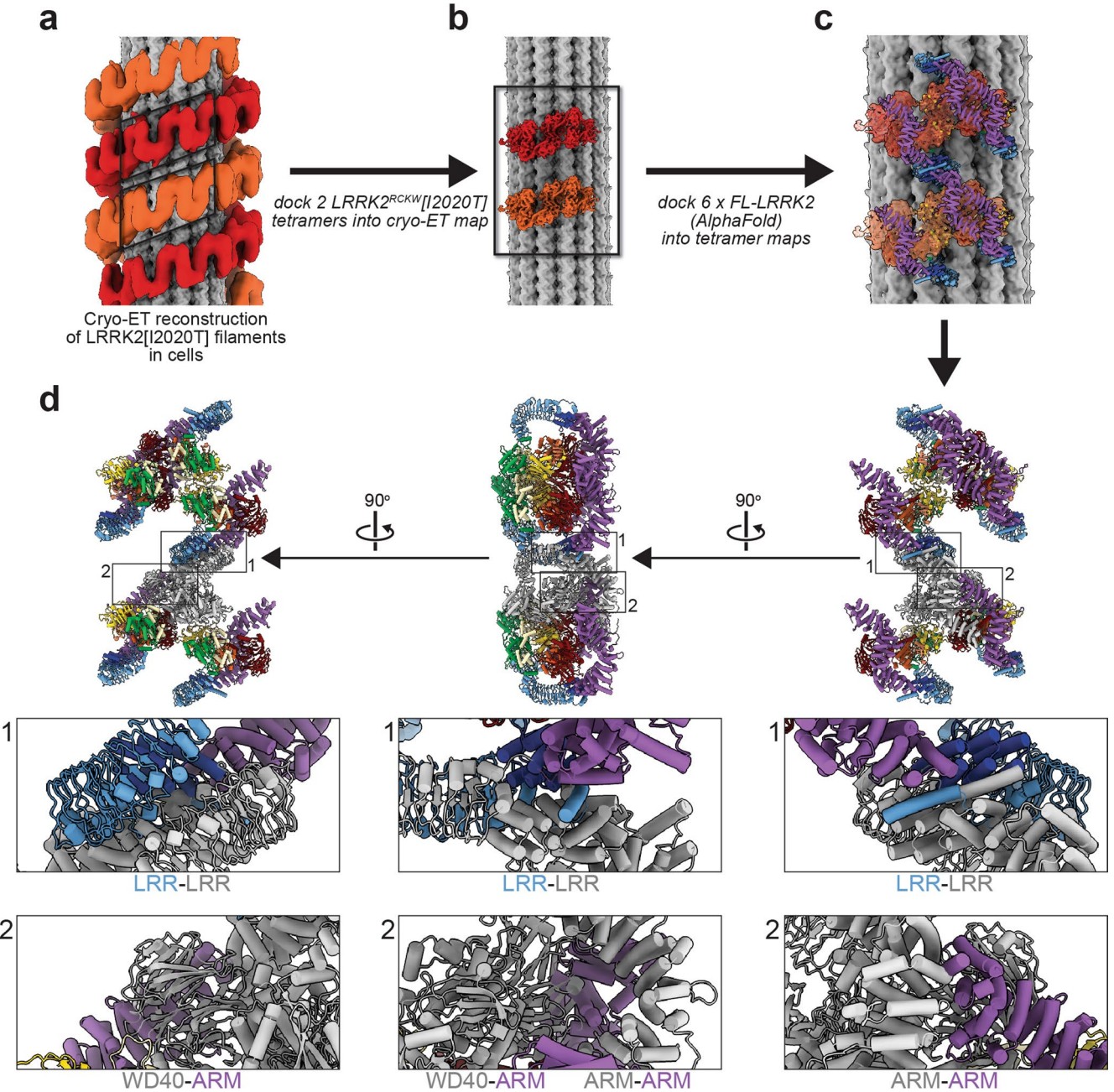

**Extended Data Fig. 6 | Modeling of full-length LRRK2 into the cryo-ET reconstruction of microtubule-associated LRRK2[I2020T] filaments in cells. a**, Cryo-ET reconstruction of microtubule-associated LRRK2[I2020T] filaments in cells[1]. The LRRK2 strands that form the double-helical filaments are shown in light and dark orange. For this figure, the density corresponding to the microtubule was replaced with a 10 Å representation of a molecular model of a microtubule. **b**, We docked copies of our 5.9 Å reconstruction of a LRRK2[RCKW][I2020T] tetramer from the microtubule-associated filaments (Fig. 1b, c) into the regions indicated by the parallelograms in (**a**). **c**, Next, we docked

two copies of the AlphaFold model of full-length LRRK2 (AF-Q5S007), which is in the active state, as is the case with LRRK2[RCKW][I2020T] in our filaments, into each of the 5.9 Å maps. The pairs of AlphaFold models in each map correspond to the COR-B:COR-B dimer. This panel shows a region corresponding to the rectangle in (**b**). **d**, Three different views of the models docked in (**c**). Below each model, close-ups show regions where adjacent filaments clash. These clashes involve a domain in the N-terminal repeats of one LRRK2, and either the same domain on another LRRK2, or the WD40 domain. For clarity, one of the LRRK2's is shown in grey instead of the standard rainbow coloring.

| | |
|---|---|

# Reporting Summary

## Statistics

For all statistical analyses, confirm that the following items are present in the figure legend, table legend, main text, or Methods section.

| n/a | Confirmed | |
|---|---|---|
| ☐ | ☒ | The exact sample size (*n*) for each experimental group/condition, given as a discrete number and unit of measurement |
| ☐ | ☒ | A statement on whether measurements were taken from distinct samples or whether the same sample was measured repeatedly |
| ☐ | ☒ | The statistical test(s) used AND whether they are one- or two-sided *Only common tests should be described solely by name; describe more complex techniques in the Methods section.* |
| ☒ | ☐ | A description of all covariates tested |
| ☐ | ☒ | A description of any assumptions or corrections, such as tests of normality and adjustment for multiple comparisons |
| ☐ | ☒ | A full description of the statistical parameters including central tendency (e.g. means) or other basic estimates (e.g. regression coefficient) AND variation (e.g. standard deviation) or associated estimates of uncertainty (e.g. confidence intervals) |
| ☐ | ☒ | For null hypothesis testing, the test statistic (e.g. *F*, *t*, *r*) with confidence intervals, effect sizes, degrees of freedom and *P* value noted *Give P values as exact values whenever suitable.* |
| ☒ | ☐ | For Bayesian analysis, information on the choice of priors and Markov chain Monte Carlo settings |
| ☒ | ☐ | For hierarchical and complex designs, identification of the appropriate level for tests and full reporting of outcomes |
| ☒ | ☐ | Estimates of effect sizes (e.g. Cohen's *d*, Pearson's *r*), indicating how they were calculated |

*Our web collection on statistics for biologists contains articles on many of the points above.*

## Software and code

Policy information about availability of computer code

| Data collection | Leginon was used for all automated EM data collection. Custom code for filament collection was used and is available at https://github.com/matyszm/filfinder. For light microscopy experiments, data was collected on commercially available Nikon Elements Software. For Western blots, data was collected using Image Studio v5.2 (Li-COR). Protein sequences of LRRK2 (Q5S007) and LRRK1 (Q38SD2) were obtained from UniProt. |
|---|---|
| Data analysis | LRRK2 on microtubule EM data was aligned with MotionCor2 and defocus was calculated with CTFIND4. Particle picking and data processing until symmetry expansion was done in Relion 3.1. After symmetry expansion all analysis was done in cryoSPARC v3.2. LRRK1 EM data was aligned with cryoSPARC patch motion and defocus was calculated with cryoSPARC's patch CTF. Particle picking was done with either cryoSPARC blob picker or crYOLO using our previously trained model from Deniston et al. All subsequent processing was done exclusively in cryoSPARC V3.2. For light microscopy experiments, ImageJ was used for analysis and to make image z-maximum projections and kymographs. Graphpad Prism v_9.2 was used for all statistical analysis of light microscopy data. For Western blot, data was quantified using ImageStudio v5.2. The version of ImageJ used was 1.53. Sequence alignments were performed with Clustal Omega web services (https://www.ebi.ac.uk/Tools/msa/clustalo/, RRID:SCR_001591) and annotated using Jalview (version 2.11, http://www.jalview.org/, RRID:SCR_006459). Data visualization and statistical analyses were performed in GraphPad Prism (version 9.2, https://www.graphpad.com/, RRID:SCR_002798) and ImageJ83 (version 1.53, https://imagej.nih.gov/ij/, RRID:SCR_003070). |

For manuscripts utilizing custom algorithms or software that are central to the research but not yet described in published literature, software must be made available to editors and reviewers. We strongly encourage code deposition in a community repository (e.g. GitHub). See the Nature Portfolio guidelines for submitting code & software for further information.

## Data

Policy information about availability of data

All manuscripts must include a data availability statement. This statement should provide the following information, where applicable:
- Accession codes, unique identifiers, or web links for publicly available datasets
- A description of any restrictions on data availability
- For clinical datasets or third party data, please ensure that the statement adheres to our policy

> All raw data that went into the biochemical and cell biological analyses were deposited in a spreadsheet with the manuscript.
>
> EM Data Bank accession numbers: EMDB-25649, -25664, -25658, -25908, -25674, -25672, -25897.
> Protein Data Bank accession numbers: PDB-7THY, 7THZ.
> EMPIAR database accession numbers: EMPIAR-10925, -10924, -10921.

# Field-specific reporting

Please select the one below that is the best fit for your research. If you are not sure, read the appropriate sections before making your selection.

☒ Life sciences    ☐ Behavioural & social sciences    ☐ Ecological, evolutionary & environmental sciences

For a reference copy of the document with all sections, see nature.com/documents/nr-reporting-summary-flat.pdf

# Life sciences study design

All studies must disclose on these points even when the disclosure is negative.

| | |
|---|---|
| Sample size | For all experiments, we determined sample size following established conventions in the field. |
| Data exclusions | For fluorescent binding experiments, clear bright aggregates were excluded from the analysis. Areas where microtubules crossed over each other were also excluded. For single-molecule kinesin experiments, clear bright aggregates (less than 5% of runs) were excluded from the analysis, as these runs display longer run lengths than typical single-molecule kinesin runs (Brouhard, 2010, Methods Cell Biol). No conclusions change with the addition or exclusion of these aggregates, and we would be happy to provide the data without exclusion of aggregates if deemed necessary. There were no exclusions made from the microtubule pelleting binding assays, the results of which were consistently qualitatively similar to that of the fluorescence-based assays. |
| Replication | Single molecule experiments in Figures 2, 4, and 5 were performed with between two and four technical replicates on separate days. All cellular filament assay data from Figures 2 and 5 were quantified from at least five technical replicates (defined in Methods as a coverslip containing between 50 and 150 cells) collected across experiments performed on at least three different days. All cellular kinase assay data from Extended Data Figures 4 and 5 were quantified from at least six technical replicates collected across three independent experiments. Replicate numbers for each figure are indicated in the raw data spreadsheet deposited with the manuscript. All attempts at replication were successful. |
| Randomization | This is not relevant. We have no data involving organisms or subjects that would require randomization. |
| Blinding | For cell biology data (LRRK2 filament assays), the experimenter was blinded to conditions both during image acquisition and analysis of LRRK2 filaments. |

# Reporting for specific materials, systems and methods

We require information from authors about some types of materials, experimental systems and methods used in many studies. Here, indicate whether each material, system or method listed is relevant to your study. If you are not sure if a list item applies to your research, read the appropriate section before selecting a response.

## Materials & experimental systems

| n/a | Involved in the study |
|---|---|
| ☐ | ☒ Antibodies |
| ☐ | ☒ Eukaryotic cell lines |
| ☒ | ☐ Palaeontology and archaeology |
| ☒ | ☐ Animals and other organisms |
| ☒ | ☐ Human research participants |
| ☒ | ☐ Clinical data |
| ☒ | ☐ Dual use research of concern |

## Methods

| n/a | Involved in the study |
|---|---|
| ☒ | ☐ ChIP-seq |
| ☒ | ☐ Flow cytometry |
| ☒ | ☐ MRI-based neuroimaging |

# Antibodies

| Antibodies used | recombinant rabbit monoclonal anti-Rab8a pT72 (abcam, ab230260, MJF-R20, Lot: GR3216587-1); rabbit anti-Rab10 pT73 (abcam, ab230261, MJF-R21, Lot: GR3216588-1); rabbit anti-LRRK2 (abcam, ab133474, c41-2, Lot: G3240365-3); chicken anti-GFP (AvesLabs, Cat: GFP-1020, Lot: GFP879484); rabbit anti-GAPDH (Cell Signaling Technology, Cat# 2188, 14C10, Lot 14-2118S); mouse anti-GFP (Santa Cruz, clone: B-2, Cat: sc-9996, Lot: C1518); AlexaFluor488 goat anti chicken IgG (Invitrogen, Ref: A11039, Lot: 1937504 2180688); IRDye 800CW goat anti-rabbit (LiCOR, P/N:; 926-32211, Lot: C90229-05); IRDye 680RD goat anti-mouse (LiCOR, P/N: 926-68070, Lot: C90219-05) |
|---|---|
| Validation | All antibodies used are well-validated and highly specific commercially available antibodies. For LiCOR quantification, linear range was determined for each antibody. The recombinant rabbit monoclonal anti-Rab8a pT72 and rabbit anti-Rab10 pT73 antibodies was validated for western blot by abcam (reference PMID: 29127256). The rabbit anti-LRRK2 antibody was validated for western blot and immunofluorescence by abcam (reference PMID: 23560750). The chicken anti-GFP antibody was validated for immunofluorescence by Aves labs, they also reference 49 instances of its use in the literature. The rabbit anti-GAPDH antibody was validated for western blot by Cell Signaling Technology, they reference 5776 instances of its use in the literature. The mouse anti-GFP antibody was validated for western blot by Santa Cruz Biotechnology, they reference 3027 instances of its use in the literature. |

# Eukaryotic cell lines

Policy information about cell lines

| Cell line source(s) | HEK293T used were from ATCC (CRL-3216). Sf9 cells (catalog number 11496015) obtained from Thermo Fisher. |
|---|---|
| Authentication | ATCC uses morphology, karyotyping, and PCR based approaches to confirm the identity of human cell lines and to rule out both intra- and interspecies contamination. These include an assay to detect species specific variants of the cytochrome C oxidase I gene (COI analysis) to rule out inter-species contamination and short tandem repeat (STR) profiling to distinguish between individual human cell lines and rule out intra-species contamination. Sf9 cells were used for protein expression and were not validated. |
| Mycoplasma contamination | New cell lines received by our lab are tested for mycoplasma before expanding and freezing. After thawing, each cell line is tested again. Every three months, all cells growing in the lab are tested for mycoplasma as well. The cells used in our experiments were last tested on 06/23/21 and did not contain contamination. |
| Commonly misidentified lines (See ICLAC register) | No commonly misidentified cell lines were used. |

