## [Peer Review File · Nature Structural & Molecular Biology]

Peer Review Information

Manuscript Title: Structural basis for Parkinson's Disease-linked LRRK2's binding to microtubules

Corresponding author name(s): Andres Leschziner, Samara Reck-Peterson

Editorial Notes:

Reviewer Comments & Decisions:

Decision Letter, initial version:

Message: 1st Mar 2022

Dear Sam,

Thank you again for submitting your manuscript "Structural basis for Parkinson's Disease-linked LRRK2's binding to microtubules". I apologize for the delay in responding, which resulted from the difficulty in obtaining suitable referee reports. Nevertheless, we now have comments (below) from the 3 reviewers who evaluated your paper. In light of those reports, we remain interested in your study and would like to see your response to the comments of the referees, in the form of a revised manuscript.

You will see that the reviewers are positive about the interest and quality of the study, but make constructive suggestions for improving it, particularly with regard to the presentation of the findings and their broader implications. Please be sure to address/respond to all concerns of the referees in full in a point-by-point response and highlight all changes in the revised manuscript text file. If you have comments that are intended for editors only, please include those in a separate cover letter.

We expect to see your revised manuscript within 6 weeks. If you cannot send it within this time, please contact us to discuss an extension; we would still consider your revision, provided that no similar work has been accepted for publication at NSMB or published elsewhere.

Reporting Summary:

When submitting the revised version of your manuscript, please pay close attention to our [Digital Image Integrity Guidelines](https://www.nature.com/nature-research/editorial-policies/image-integrity).

Data availability: this journal strongly supports public availability of data. All data used in accepted papers should be available via a public data repository, or alternatively, as Supplementary Information. If data can only be shared on request, please explain why in your Data Availability Statement, and also in the correspondence with your editor. Please note that for some data types, deposition in a public repository is mandatory - more information on our data deposition policies and available repositories can be found below: <https://www.nature.com/nature-research/editorial-policies/reporting-standards#availability-of-data>

We require deposition of coordinates (and, in the case of crystal structures, structure factors) into the Protein Data Bank with the designation of immediate release upon publication (HPUB). Electron microscopy-derived density maps and coordinate data must be deposited in EMDB and released upon publication. To avoid delays in publication, dataset accession numbers must be supplied with the final accepted manuscript and appropriate release dates must be indicated at the galley proof stage.

[Redacted]

Kind regards,
Florian

Florian Ullrich, Ph.D.
Associate Editor
Nature Structural & Molecular Biology
ORCID 0000-0002-1153-2040

Referee expertise:

Referee #1: LRRK2 function

Referee #2: structural biology, LRRK2

Referee #3: structural biology, cryo-EM, microtubules

Reviewers' Comments:

Reviewer #1:

Remarks to the Author:

This is an elegant study in which the authors provide the highest resolution structures of the catalytic moiety of LRRK2 reported to date, in the active closed conformation. The

study provides a wealth of other important information that will make a significant contribution to the LRRK2 research field. This includes identifying the key highly conserved residues within the LRRK2 ROC GTPase domain that control interactions with microtubules. These mutants will be invaluable at enabling researchers to better probe the physiological roles that LRRK2 binding to microtubules plays, and study whether this is implicated in the pathogenesis of Parkinson's disease. The authors also characterise key residues in the WD40 and CORB domains that mediate dimer interphases. Interestingly, they demonstrate that mutation of residues R1731 in the CORB domain reduces binding to microtubules and also enhances LRRK2 activity towards Rab proteins. A double CORB-WD40 domain mutant that is unable to form dimers no longer interacted with microtubules or inhibited kinesin motility. This data demonstrates conclusively that the dimeric form is required for interaction with microtubules. In addition, the authors provide the first description of the structure of the catalytic moiety of LRRK1 in the inactive conformation, revealing that its structure is overall quite like LRRK2. The authors' finding that LRRK1 does not interact with microtubules and possesses a similar overall domain structure for LRRK2 is of interest. This study will help researchers studying LRRK1 biology. I would be strongly in favour of this study being accepted for publication in NSMB. Below I suggest some minor points the authors might wish to consider in a revised version.

1. A lot of the interesting data in this paper is presented in the extended data sections rather than in the main figures. I would recommend that the authors go through the study and consider including some of the data currently presented in the extended data section in the main paper instead. In my opinion most of the data presented in extended data figs.4 & 5 would be suitable to present in the main figures.
2. Many of the structural figures in the main figures (figs.1, 2A and 2, 3A, 4A, 4C, 4D, 5A, 5B & 5C) are quite small. Clarity would be enhanced if the figures were made significantly larger in my opinion.
3. Does the authors' data suggest that the fully active conformation of LRRK2 is a tetramer rather than a dimer? Potentially the discussion surrounding this could be expanded a bit more. Is there any biophysical evidence that active LRRK2 forms a tetramer?
4. Do the authors have a hypothesis of why the R1731 mutations that disrupt the CORB dimer interphase enhance LRRK2 mediated phosphorylation of Rab proteins? Presumably such mutants would abolish the ability of LRRK2 to form dimers/tetramers and this might be predicted to inhibit Rab protein phosphorylation if dimer formation was important for activation?
5. The authors explore the impact that the double R1731D/S2345D mutant has on binding to microtubules. Have the authors explored how this double mutant impacts the ability of LRRK2 to phosphorylate Rab proteins in vitro or in vivo?
6. The data shown in extended fig.4Q that microtubule binding of LRRK2 is salt dependent seems important. As 150 millimolar physiological salt significantly reduces microtubule binding, can the authors discuss on whether the physiological salt concentrations in vivo would be expected to significantly suppress LRRK2 from binding microtubules?
7. For the experiment shown in fig.3E involving the removal of the tubulin tail with the subtilisin protease, is a control needed to demonstrate that the protease has substantially

cleaved the carboxy terminus of tubulin tails in the experiment performed?

8. I presume that the resolution of the structures analysed does not permit visualisation of how MLI2 interacts with LRRK2. For the non-expert this could be potentially mentioned in the text.

Reviewer #2:

Remarks to the Author:

A. Summary of the key results.

This manuscript titled "Structural basis for Parkinson's Disease-linked LRRK2's binding to microtubules" by a strong research team, that includes one of the pioneers in LRRK2 structure determination, provides important insight into the basis of LRRK2 binding to microtubules. Certainly, this manuscript is an important step forward relative to the low resolution structures of LRRK2 bound to microtubules currently available. A comprehensive and sound study that combines cryo-EM with biochemical and biophysical assays, the data presented provides new details on the interactions and mode of binding of LRRK2 to microtubules, the effect of certain type of inhibitors on this binding, and the differences with the analogous region in LRRK1, which lacks the ability to bind to microtubules and is not involved in Parkinson's Disease.

B. Originality and significance.

The biological and pathological relevance of LRRK2 binding to microtubules remains to be determined, and this is generally speaking the Achilles heel of the manuscript, however the structural and biophysical insight provided will likely help examine the relevance of the LRRK2-microtubule interaction, as well as open new possible structural target sites for drug discovery.

The manuscript presents very important data to advance the field of LRRK2 and Parkinson's disease and should be interesting for a broad community of structural biologists, cell biologists and neuroscientists.

C. Data & methodology.

From a cryo-EM perspective, the work is impressive. The combination of helical diffraction and single particle approaches must have been challenging and the success presented here is commendable, and will probably inspire other laboratories to apply similar approaches to comparable systems. The methods section is extensive and contains important details, and the supplementary figures bring clarity to explain the elaborate cryo-EM data processing strategy.

D. Appropriate use of statistics and treatment of uncertainties.

Data statistics and analysis are sound.

E. Suggested improvements: experiments, data for possible revision.

The authors should check for directional anisotropy of the pertinent reconstructions and

include directional FSC curves (for instance those provided by using <https://3dfsc.salk.edu/>).

In general the figures lack information about the dimensions of the objects, for instance Fig. 1b should display the dimensions of the tube (or at least of the rhomboid selection), and scale bars are conspicuously absent from all microscopy (whether electron or fluorescence) images. Cryo-EM class averages should display them too.

F. References.

The references are appropriate and the manuscript gives due credit to previous work.

G. Clarity and context.

The manuscript is clearly written with appropriate figures. However, the text is on the long side and the discussion could be shortened significantly, as considerable sections of it are to some extent a recapitulation of the results section.

Reviewer #3:

Remarks to the Author:

Mutations in Leucine Rich Repeat Kinase 2 (LRRK2) are common among familial cases of Parkinson's disease (PD), and are also linked to sporadic PD cases. Insights into LRRK2 structure and function are thus important in understanding disease mechanisms and identifying routes for therapy development. LRRK2 is a large, multi-domain protein containing a number of protein interaction modules as well as a kinase domain. All disease-linked mutations so far identified increase kinase activity. Although the function of LRRK2 is not precisely understood, current evidence points to role(s) in membrane trafficking. LRRK2 can also interact with microtubules, LRRK2 over-expression induces extended LRRK2 filaments on cellular microtubules and many PD linked mutations increase association of LRRK2 with microtubules in cells. Kinase activity and microtubule binding are also linked, as evidenced, for example, by different modes of kinase inhibition having different effects on LRRK2 microtubule binding and consequent effects on microtubule-based-motor movement: type I inhibitors stabilise the "closed" conformation of the kinase domain, which also promotes microtubule binding and blocks motor movement, while type II inhibitors have the opposite effects.

Recent studies, including by the authors of the current work, have provided a number of key insights about LRRK2 structure/function. The structures of full length LRRK2 as well as its catalytic half were recently determined (Myasnikov et al, Deniston et al), while in situ cryo-electron tomography revealed medium resolution structure of microtubule-associated filaments formed by LRRK2 with the PD mutation I2020T (Watanabe et al). This enabled derivation of a model of microtubule-bound LRRK2 and revealed crucial insight about the complex sets of inter-protein interactions in the C-terminal part of the protein (referred to as LRRK2-RCKW in the current study) that stabilise LRRK2 filament formation on microtubules (Watanabe et al). In Deniston et al, similar filaments were also reported on microtubules in vitro. The symmetry of the LRRK2 filaments does not match the left-handed symmetry of the underlying microtubule, hinting that the microtubules may act as a relatively non-specific scaffold for LRRK2 filament formation – probably mediated by their charge (as predicted in Extended Data Figure 7k by Deniston et al) – rather than

acting as a precise template. The interactions between LRRK2 via its ROC domain and the microtubule could not be visualised, and this was hypothesised to be due to flexibility at that interface. Furthermore, there are no common PD-linked mutations within the charged region of LRRK2 ROC domain.

The study of Snead et al is a logical extension of this previous work. The authors describe optimization of conditions for more ordered interaction of LRRK2-RCKW with microtubules, and use cryo-EM and single particle averaging to obtain a higher resolution view of the LRRK2-LRRK2 and LRRK2-microtubule interactions than previously available. The best ordered filaments were formed by LRRK2-RCKW[I2020T] + MLI-2, a type I inhibitor, bound to non-canonical 11/12 protofilament microtubules. Less well-ordered arrays were also formed by wild-type LRRK2-RCKW and the G2019S mutant, and these in vitro filaments were described as similar although not identical to those observed in cells.

Density corresponding to the LRRK2-RCKW ROC domain is now visible in the Snead et al structures. Because of the mismatch between LRRK2 filament and microtubule symmetry, the pseudo-2-fold symmetry of the LRRK2 filaments reveals two different modes of interaction between the ROC domain and the microtubule surface, although the same highly charged surface of ROC is involved in both. Consistently, the LRRK2-RCKW interaction with microtubules is salt-sensitive and is mediated at least in part by charged, unstructured tubulin C-terminal tails. Furthermore, the authors also determine the structure of the related LRRK1-RCKW which, although exhibiting an overall similar structure to LRRK2-RCKW, does not bind microtubules. Although arguably, sequence comparison alone would have been sufficient to draw this conclusion, the structural comparison between the related proteins thus further reinforces the importance of charge mediated microtubule interaction by the LRRK2 ROC domain.

Previous data were consistent with a “closed” conformation of the kinase in supporting filament formation, and the higher resolution structure now confirms this and shows more details of that conformation. The higher resolution achieved also allows more precise mapping of LRRK2-LRRK2 domain interactions – WD40:WD40 and COR-B:COR-B interfaces – in the filaments. The authors use mutagenesis to disrupt interface interactions and test effects on filament formation in cells, together with microtubule binding and inhibition of microtubule-based motor movement.

Overall the data are solid, the manuscript is well written, and describes technically rigorous and thorough work that validates previous ideas about the properties of this important protein. These findings will be very useful for those working in PD research. However, it is not obvious what the major new advances are in terms of key open questions relating to LRRK2 function, the correlation between kinase activity and microtubule binding, and the specific link between PD disease mechanisms and LRRK2 microtubule binding. This link is rendered particularly uncertain by the data presented in (Fig 5h, Extended Data Fig. 5k) which shows that the disease-variant mutant R1501W reduces LRRK2 microtubule binding.

Below are some further suggestions that would improve the manuscript.

1. Watanabe et al proposed that LRRK2 N-terminal domains might be involved in maintaining the spacing between the full-length LRRK2 filaments on the microtubule surface. Although the pitch of the filaments formed by the LRRK2-RCKW is similar to that of the filaments formed by full length LRRK2, the formation of a triple (LRRK2-RCKW)

versus double (full-length) helix suggests filament organisation in vitro might not be fully representative of the physiological state in the cell. It would therefore be useful to provide a comparison of the overall fold of the domains fit into both structures and show the similarities/changes in assembly of the filaments, including positions along the microtubule lattice.

2. The C-terminal construct is defined twice – lines 78 and line 112

3. The type I kinase inhibitor MLi-2 is not defined when first introduced in the main text (line 142)

4. Providing values for helical spacing indicated by the layer lines in Extended Data Figure 1a will help understand and compare organisation of different LRRK2 filaments on the microtubule.

Author Rebuttal to Initial comments

1st Mar 2022

Dear Sam,

Thank you again for submitting your manuscript "Structural basis for Parkinson's Disease-linked LRRK2's binding to microtubules". I apologize for the delay in responding, which resulted from the difficulty in obtaining suitable referee reports. Nevertheless, we now have comments (below) from the 3 reviewers who evaluated your paper. In light of those reports, we remain interested in your study and would like to see your response to the comments of the referees, in the form of a revised manuscript.

You will see that the reviewers are positive about the interest and quality of the study, but make constructive suggestions for improving it, particularly with regard to the presentation of the findings and their broader implications. Please be sure to address/respond to all concerns of the referees in full in a point-by-point response and highlight all changes in the revised manuscript text file. If you have comments that are intended for editors only, please include those in a separate cover letter.

We expect to see your revised manuscript within 6 weeks. If you cannot send it within this time, please contact us to discuss an extension; we would still consider your revision, provided that no similar work has been accepted for publication at NSMB or published elsewhere.

Reporting Summary:

When submitting the revised version of your manuscript, please pay close attention to our <https://www.nature.com/nature-research/editorial-policies/image-integrity>>Digital Image Integrity Guidelines.

Data availability: this journal strongly supports public availability of data. All data used in accepted papers should be available via a public data repository, or alternatively, as Supplementary Information. If data can only be shared on request, please explain why in your Data Availability Statement, and also in the correspondence with your editor. Please note that for some data types, deposition in a public repository is mandatory - more information on our data deposition policies and available repositories can be found below:

<https://www.nature.com/nature-research/editorial-policies/reporting-standards#availability-of-data>

We require deposition of coordinates (and, in the case of crystal structures, structure factors) into the Protein Data Bank with the designation of immediate release upon publication (HPUB). Electron microscopy-derived density maps and coordinate data must be deposited in EMDB and released upon publication. To avoid delays in publication, dataset accession numbers must be supplied with the final accepted manuscript and appropriate release dates must be indicated at the galley proof stage.

Nature Structural & Molecular Biology is committed to improving transparency in authorship. As part of our efforts in this direction, we are now requesting that all authors identified as 'corresponding author' on published papers create and link their Open Researcher and Contributor Identifier (ORCID) with their account on the Manuscript Tracking System (MTS), prior to acceptance. This applies to primary research papers only. ORCID helps the scientific community achieve unambiguous attribution of all scholarly contributions. You can create and link your ORCID from the home page of the MTS by clicking on 'Modify my Springer Nature account'. For more information please visit please visit www.springernature.com/orcid.

[Redacted]

Kind regards,
Florian

Florian Ullrich, Ph.D.
Associate Editor
Nature Structural & Molecular Biology
ORCID 0000-0002-1153-2040

Referee expertise:

Referee #1: LRRK2 function

Referee #2: structural biology, LRRK2

Referee #3: structural biology, cryo-EM, microtubules

Reviewers' Comments:

Reviewer #1:

Remarks to the Author:

This is an elegant study in which the authors provide the highest resolution structures of the catalytic moiety of LRRK2 reported to date, in the active closed conformation. The study provides a wealth of other important information that will make a significant contribution to the LRRK2 research field. This includes identifying the key highly conserved residues within the LRRK2 ROC GTPase domain that control interactions with microtubules. These mutants will be invaluable at enabling researchers to better probe the physiological roles that LRRK2 binding to microtubules plays, and study whether this is implicated in the pathogenesis of Parkinson's disease. The authors also characterise key residues in the WD40 and CORB domains that mediate dimer interphases. Interestingly, they demonstrate that mutation of residues R1731 in the CORB domain reduces binding to microtubules and also enhances LRRK2 activity towards Rab proteins. A double CORB-WD40 domain mutant that is unable to form dimers no longer interacted with microtubules or inhibited kinesin motility. This data demonstrates conclusively that the dimeric form is required for interaction with microtubules. In addition, the authors provide the first description of the structure of the catalytic moiety of LRRK1 in the inactive conformation, revealing that its structure is overall quite like LRRK2. The authors' finding that LRRK1 does not interact with microtubules and possesses a similar overall domain structure for LRRK2 is of interest. This study will help researchers studying LRRK1 biology.

I would be strongly in favour of this study being accepted for publication in NSMB. Below I suggest some minor points the authors might wish to consider in a revised version.

We would like to thank the reviewer for the very positive comments on our work.

We wanted to clarify a couple of points based on the remarks above.

- Although the CORB and WD40 mutants were designed to disrupt interfaces, we have no data directly showing that the double mutant does not form dimers. That would require running size exclusion chromatography with high concentrations of the protein. The challenge lies in figuring out what concentrations in solution are equivalent to the effective concentrations of LRRK2 once it is constrained on the microtubule surface. There, the protein is limited to a 2D surface with a dramatic decrease in the degrees of rotational freedom. In fact, cryo-EM imaging with S2345D (EDF4h) shows that the mutant still forms filaments, although these are less ordered than WT. This suggests that the WD40-WD40 interface can still form, at least at the very high concentrations used in these experiments and under the constraints imposed by the microtubule.
- We refrained from stating that a dimer is required for interaction with microtubules because that would imply that a monomer is incapable of binding to microtubules at any concentration, something we did not (and could not, in practical terms) test. Our data do show that, at the concentrations tested, a dimer is the smallest species where binding to microtubules could be detected. We have no reason to believe that a monomer would be incapable of interacting with the microtubule, but the avidity effects coming from the dimer would likely mean that the affinity of the monomer-microtubule interaction would be significantly weaker (and thus challenging to detect/measure experimentally).

1. A lot of the interesting data in this paper is presented in the extended data sections rather than in the main figures. I would recommend that the authors go through the study and consider including some of the data currently presented in the extended data section in the main paper instead. In my opinion most of the data presented in extended data figs.4 & 5 would be suitable to present in the main figures.

We agree with the reviewer's assessment; we were overly conservative with the data we presented in the main figures even when we thought it belonged there. We have now taken the reviewer's advice and have moved some of the data from Extended Data Figs. 4 and 5 back into main Figures 2, 4, and 5.

2. Many of the structural figures in the main figures (figs.1, 2A and 2, 3A, 4A, 4C, 4D, 5A, 5B & 5C) are quite small. Clarity would be enhanced if the figures were made significantly larger in my opinion.

We thank the reviewer for the suggestion and have enlarged all the suggested structures.

3. Does the authors' data suggest that the fully active conformation of LRRK2 is a tetramer rather than a dimer? Potentially the discussion surrounding this could be expanded a bit more. Is there any biophysical evidence that active LRRK2 forms a tetramer?

We do not think our data suggest this and are not aware of any biophysical evidence that LRRK2 forms tetramers in solution (at least at physiological concentrations). Our data show a statistically significant increase in the phosphorylation of Rab10 in cells for the R1731D mutant, designed to disrupt the CORB interface, which is the interface observed in the structure of the dimeric form of full-length LRRK2. If anything, this data points to the dimer as being a less active form of LRRK2, with disruption of this interaction leading to activation. We do not know at this point whether the activation relates directly to the state of the protein as a monomer or a dimer (there is no obvious structural explanation as to why this would be the case), or whether it reflects changes in subcellular localization. We now mention this in the Discussion.

4. Do the authors have a hypothesis of why the R1731 mutations that disrupt the CORB dimer interphase enhance LRRK2 mediated phosphorylation of Rab proteins? Presumably such mutants would abolish the ability of LRRK2 to form dimers/tetramers and this might be predicted to inhibit Rab protein phosphorylation if dimer formation was important for activation?

We agree with the reviewer that this is an intriguing piece of data. As mentioned above, there is no obvious structural explanation as to why disruption of the COR-B dimerization interface would activate LRRK2's kinase activity. Current data show no differences in the structure of LRRK2 in its monomeric and dimeric forms; dimerization does not stabilize the autoinhibited state of LRRK2 in any obvious way. Consequently, we would not have predicted that breaking the dimerization interface would activate LRRK2's kinase. However, since the Rab10 phosphorylation assays were performed in cells, our hypothesis is that the shift from dimer to monomers caused by the R1731D mutation is changing the cellular localization of LRRK2. This could then lead to increased Rab10 phosphorylation due to location-dependent activation of LRRK2 and/or increased proximity to Rab10.

We have now moved the data on Rab10 phosphorylation by the different dimerization interface mutants from the Extended Data to the main Figures, as suggested by the reviewer, and bring up the hypothesis mentioned above in the Discussion.

5. The authors explore the impact that the double R1731D/S2345D mutant has on binding to microtubules. Have the authors explored how this double mutant impacts the ability of LRRK2 to phosphorylate Rab proteins in vitro or in vivo?

We did not explore Rab10 phosphorylation for the R1731D/S2345D double mutant based on our results with the single mutants (R1731D and S2345D). Although the results of Rab10 phosphorylation in cells with the single mutants turned out to be more interesting than we had anticipated (as discussed above), these experiments were done to test whether the mutations we had introduced affected the structure of the protein. (We would have taken a decrease in Rab10 phosphorylation with any mutant as a warning sign that the mutation might have affected more than the interface we meant to disrupt.) The fact that neither single mutant had resulted in a decrease in Rab10 phosphorylation indicated that the mutations had not disrupted the structure of the protein; we thus felt that testing the double mutant was not necessary.

6. The data shown in extended fig.4Q that microtubule binding of LRRK2 is salt dependent seems important. As 150 millimolar physiological salt significantly reduces microtubule binding, can the authors discuss on whether the physiological salt concentrations in vivo would be expected to significantly suppress LRRK2 from binding microtubules?

The data in Extended Data Fig. 4I-n (new numbering) as well as Fig.3d all point to weaker interactions between LRRK2 and microtubules at physiological salt concentrations, as would be expected given the electrostatic nature of the interface. In this sense, we do not think of physiological salt concentrations as "suppressing" LRRK2 binding to microtubules, but rather as revealing that the relevant microtubule-bound species is likely to be a short LRRK2 oligomer. The non-physiological low salt concentrations are simply a useful experimental tool to enhance the electrostatic interaction between LRRK2 and the microtubule for detection purposes. This is one of the reasons why we propose in the Discussion that the most likely species of microtubule-associated LRRK2 in cells would be short oligomers (dimers, tetramers...) instead of the extended filaments seen under overexpression conditions or with low salt concentrations.

7. For the experiment shown in fig.3E involving the removal of the tubulin tail with the subtilisin

protease, is a control needed to demonstrate that the protease has substantially cleaved the carboxy terminus of tubulin tails in the experiment performed?

We thank the reviewer for pointing out this omission. These data are now shown in Extended Data Fig.4o.

8. I presume that the resolution of the structures analysed does not permit visualisation of how MLI2 interacts with LRRK2. For the non-expert this could be potentially mentioned in the text.

Correct. This is a good suggestion and we now mention this in the text.

Reviewer #2:

Remarks to the Author:

A. Summary of the key results.

This manuscript titled “Structural basis for Parkinson’s Disease-linked LRRK2’s binding to microtubules” by a strong research team, that includes one of the pioneers in LRRK2 structure determination, provides important insight into the basis of LRRK2 binding to microtubules. Certainly, this manuscript is an important step forward relative to the low resolution structures of LRRK2 bound to microtubules currently available. A comprehensive and sound study that combines cryo-EM with biochemical and biophysical assays, the data presented provides new details on the interactions and mode of binding of LRRK2 to microtubules, the effect of certain type of inhibitors on this binding, and the differences with the analogous region in LRRK1, which lacks the ability to bind to microtubules and is not involved in Parkinson’s Disease.

B. Originality and significance.

The biological and pathological relevance of LRRK2 binding to microtubules remains to be determined, and this is generally speaking the Achilles heel of the manuscript, however the structural and biophysical insight provided will likely help examine the relevance of the LRRK2-microtubule interaction, as well as open new possible structural target sites for drug discovery.

The manuscript presents very important data to advance the field of LRRK2 and Parkinson’s disease and should be interesting for a broad community of structural biologists, cell biologists and neuroscientists.

C. Data & methodology.

From a cryo-EM perspective, the work is impressive. The combination of helical diffraction and single particle approaches must have been challenging and the success presented here is commendable, and will probably inspire other laboratories to apply similar approaches to comparable systems. The methods section is extensive and contains important details, and the supplementary figures bring clarity to explain the elaborate cryo-EM data processing strategy.

D. Appropriate use of statistics and treatment of uncertainties.

Data statistics and analysis are sound.

E. Suggested improvements: experiments, data for possible revision.

The authors should check for directional anisotropy of the pertinent reconstructions and include directional FSC curves (for instance those provided by using <https://3dfsc.salk.edu/>).

Directional FSC curves are now shown for all the reconstructions discussed in the text.

In general the figures lack information about the dimensions of the objects, for instance Fig. 1b should display the dimensions of the tube (or at least of the rhomboid selection), and scale bars are conspicuously absent from all microscopy (whether electron or fluorescence) images. Cryo-EM class averages should display them too.

We have added scale bars to several figures: Fig. 1b; Extended Data Fig. 1a; Extended Data Fig. 4a,b,f,h; and Extended Data Fig. 5a,f-h.

F. References.

The references are appropriate and the manuscript gives due credit to previous work.

G. Clarity and context.

The manuscript is clearly written with appropriate figures. However, the text is on the long side and the discussion could be shortened significantly, as considerable sections of it are to some extent a recapitulation of the results section.

We have shrunk the Discussion as suggested. There is one small addition to it in the revised version, in response to a comment from Reviewer 1.

Reviewer #3:

Remarks to the Author:

Mutations in Leucine Rich Repeat Kinase 2 (LRRK2) are common among familial cases of Parkinson's disease (PD), and are also linked to sporadic PD cases. Insights into LRRK2 structure and function are thus important in understanding disease mechanisms and identifying routes for therapy development. LRRK2 is a large, multi-domain protein containing a number of protein interaction modules as well as a kinase domain. All disease-linked mutations so far identified increase kinase activity. Although the function of LRRK2 is not precisely understood, current evidence points to role(s) in membrane trafficking. LRRK2 can also interact with microtubules, LRRK2 over-expression induces extended LRRK2 filaments on cellular microtubules and many PD linked mutations increase association of LRRK2 with microtubules in cells. Kinase activity and microtubule binding are also linked, as evidenced, for example, by different modes of kinase inhibition having different effects on LRRK2 microtubule

binding and consequent effects on microtubule-based-motor movement: type I inhibitors stabilise the "closed" conformation of the kinase domain, which also promotes microtubule binding and blocks motor movement, while type II inhibitors have the opposite effects.

Recent studies, including by the authors of the current work, have provided a number of key insights about LRRK2 structure/function. The structures of full length LRRK2 as well as its catalytic half were recently determined (Myasnikov et al, Deniston et al), while in situ cryo-electron tomography revealed medium resolution structure of microtubule-associated filaments formed by LRRK2 with the PD mutation I2020T (Watanabe et al). This enabled derivation of a model of microtubule-bound LRRK2 and revealed crucial insight about the complex sets of inter-protein interactions in the C-terminal part

of the protein (referred to as LRRK2-RCKW in the current study) that stabilise LRRK2 filament formation on microtubules (Watanabe et al). In Deniston et al, similar filaments were also reported on microtubules in vitro. The symmetry of the LRRK2 filaments does not match the left-handed symmetry of the underlying microtubule, hinting that the microtubules may act as a relatively non-specific scaffold for LRRK2 filament formation – probably mediated by their charge (as predicted in Extended Data Figure 7k by Deniston et al) – rather than acting as a precise template. The interactions between LRRK2 via its ROC domain and the microtubule could not be visualised, and this was hypothesised to be due to flexibility at that interface. Furthermore, there are no common PD-linked mutations within the charged region of LRRK2 ROC domain.

The study of Snead et al is a logical extension of this previous work. The authors describe optimization of conditions for more ordered interaction of LRRK2-RCKW with microtubules, and use cryo-EM and single particle averaging to obtain a higher resolution view of the LRRK2-LRRK2 and LRRK2-microtubule interactions than previously available. The best ordered filaments were formed by LRRK2-RCKW[I2020T] + MLI-2, a type I inhibitor, bound to non-canonical 11/12 protofilament microtubules. Less well-ordered arrays were also formed by wild-type LRRK2-RCKW and the G2019S mutant, and these in vitro filaments were described as similar although not identical to those observed in cells.

Density corresponding to the LRRK2-RCKW ROC domain is now visible in the Snead et al structures. Because of the mismatch between LRRK2 filament and microtubule symmetry, the pseudo-2-fold symmetry of the LRRK2 filaments reveals two different modes of interaction between the ROC domain and the microtubule surface, although the same highly charged surface of ROC is involved in both. Consistently, the LRRK2-RCKW interaction with microtubules is salt-sensitive and is mediated at least in part by charged, unstructured tubulin C-terminal tails. Furthermore, the authors also determine the structure of the related LRRK1-RCKW which, although exhibiting an overall similar structure to LRRK2-RCKW, does not bind microtubules. Although arguably, sequence comparison alone would have been sufficient to draw this conclusion, the structural comparison between the related proteins thus further reinforces the importance of charge mediated microtubule interaction by the LRRK2 ROC domain.

Previous data were consistent with a “closed” conformation of the kinase in supporting filament formation, and the higher resolution structure now confirms this and shows more details of that conformation. The higher resolution achieved also allows more precise mapping of LRRK2-LRRK2 domain interactions – WD40:WD40 and COR-B:COR-B interfaces – in the filaments. The authors use mutagenesis to disrupt interface interactions and test effects on filament formation in cells, together with microtubule binding and inhibition of microtubule-based motor movement.

Overall the data are solid, the manuscript is well written, and describes technically rigorous and thorough work that validates previous ideas about the properties of this important protein. These findings will be very useful for those working in PD research. However, it is not obvious what the major new advances are in terms of key open questions relating to LRRK2 function, the correlation between kinase activity and microtubule binding, and the specific link between PD disease mechanisms and LRRK2 microtubule binding. This link is rendered particularly uncertain by the data presented in (Fig 5h, Extended Data Fig. 5k) which shows that the disease-variant mutant R1501W reduces LRRK2 microtubule binding.

Below are some further suggestions that would improve the manuscript.

1. Watanabe et al proposed that LRRK2 N-terminal domains might be involved in maintaining the

spacing between the full-length LRRK2 filaments on the microtubule surface. Although the pitch of the filaments formed by the LRRK2-RCKW is similar to that of the filaments formed by full length LRRK2, the formation of a triple (LRRK2-RCKW) versus double (full-length) helix suggests filament organisation in vitro might not be fully representative of the physiological state in the cell. It would therefore be useful to provide a comparison of the overall fold of the domains fit into both structures and show the similarities/changes in assembly of the filaments, including positions along the microtubule lattice.

Although this is an interesting and important point, it is not possible to make this comparison in a meaningful way given the very different resolutions and modeling approaches used in the Watanabe et al. paper and in our work.

Watanabe and colleagues used Integrative Modeling to dock experimental structures or homology models of the different domains of LRRK2 into a 14Å cryo-ET map of LRRK2 filaments in cells. Their approach generated many possible solutions: 1,167 different models that accounted for the cryo-ET map were deposited. All the models had the domains in their correct general location given what we now know about LRRK2 based on subsequent higher resolution structures. However, the orientation of those domains varied significantly from model to model. In our previous paper reporting the 3.5Å structure of LRRK2^{RCKW} (Deniston, Salogiannis, Mathea et al., 2020.), we included a comparison between our structure of LRRK2^{RCKW} and all 1,167 integrative models built into the cryo-ET map of the filaments in cells (Extended Data Fig.4a in that paper). Despite the overall correct positions of the domains in the integrative models, even the one most similar to our LRRK2^{RCKW} structure had an RMSD of 10.9Å. Given this, it would not be possible to determine whether any differences between our structure of the in vitro reconstituted LRRK2^{RCKW} filaments and that from Watanabe et al. is reporting on true differences between reconstituted and cellular filaments or on the limitations of modeling into a 14Å map.

Analogous issues would arise when trying to compare the positions of LRRK2 along the microtubule lattice. The first challenge here is that Watanabe et al. focused their initial processing on the microtubule to sort out different protofilament numbers, but subsequent subtomogram averaging included both the microtubule and LRRK2. In contrast, we gave significant weight to the microtubule well into the single-particle data processing of the LRRK2^{RCKW} reconstituted filaments, resulting in a 5.4Å structure of the two protofilaments with which a LRRK2^{RCKW} dimer interacts. This means that in our structure the microtubule lattice imposed more stringent constraints on the final structure. The second challenge is again the very different resolution of the two structures. It would not be possible to accurately dock a microtubule structure into the 14Å cryo-ET map, while our 5.4Å map is sufficient to see secondary structure and establish microtubule polarity unambiguously. Thus, comparison of models built into the cryo-ET and our map are more likely to report on the different approaches used than on any underlying structural differences.

As we point out in the Discussion, we believe that the difference in strand spacing, which explains the double vs. triple helix, agrees with Watanabe et al.'s proposal that the N-terminal half of LRRK2 is involved in setting the spacing between the strands. The similar pitch, on the other hand, argues against significant differences in filament structure, as those differences would have been apparent even at the 14Å resolution of the cryo-ET map. It is likely that more subtle differences do exist between physiological and reconstituted filaments, but those are the differences that cannot be fully addressed in the absence of a higher resolution structure of LRRK2 filaments in cells.

2. The C-terminal construct is defined twice – lines 78 and line 112

Thank you for pointing this out. We have removed the second one.

3. The type I kinase inhibitor MLi-2 is not defined when first introduced in the main text (line 142)

Thank you for bringing this up. We now defined the inhibitor and provided references to its discovery and initial tests.

4. Providing values for helical spacing indicated by the layer lines in Extended Data Figure 1a will help understand and compare organisation of different LRRK2 filaments on the microtubule.

We have added this information to all the layer lines.

This email has been sent through the Springer Nature Tracking System NY-610A-NPG&MTS

Confidentiality Statement:

This e-mail is confidential and subject to copyright. Any unauthorised use or disclosure of its contents is prohibited. If you have received this email in error please notify our Manuscript Tracking System Helpdesk team at <http://platformsupport.nature.com>.

Details of the confidentiality and pre-publicity policy may be found here <http://www.nature.com/authors/policies/confidentiality.html>
Privacy Policy | Update Profile

Decision Letter, first revision:

Message: Our ref: NSMB-A45874A

17th May 2022

Dear Sam,

Thank you for submitting your revised manuscript "Structural basis for Parkinson's Disease-linked LRRK2's binding to microtubules" (NSMB-A45874A). It has now been seen again by reviewer #1 and their comments are below. Also, reviewer #3 has briefly checked the study and found it to be technically valid, but opted not to submit a full report. Because the reviewers find that the paper has improved in revision, we'll be happy in principle to publish it in Nature Structural & Molecular Biology, pending minor revisions to comply with our editorial and formatting guidelines.

To facilitate our work at this stage, we would appreciate if you could send us the main text as a word file. Please make sure to copy the NSMB account (cc'ed above).

Kind regards,
Florian

Florian Ullrich, Ph.D.
Associate Editor
Nature Structural & Molecular Biology
ORCID 0000-0002-1153-2040

Reviewer #1 (Remarks to the Author):

The authors have carefully revised their manuscript to address my comments and that of the other Reviewers in my opinion. I would recommend that the manuscript is accepted

Decision Letter, Final Checks

Our ref: NSMB-A45874A

16th Jun 2022

Dear Dr. Reck-Peterson,

Thank you for your patience as we've prepared the guidelines for final submission of your Nature Structural & Molecular Biology manuscript, "Structural basis for Parkinson's Disease-linked LRRK2's binding to microtubules" (NSMB-A45874A). Please carefully follow the step-by-step instructions provided in the attached file, and add a response in each row of the table to indicate the changes that you have made. Ensuring that each point is addressed will help to ensure that your revised manuscript can be swiftly handed over to our production team.

In recognition of the time and expertise our reviewers provide to Nature Structural & Molecular Biology's editorial process, we would like to formally acknowledge their contribution to the external peer review of your manuscript entitled "Structural basis for Parkinson's Disease-linked LRRK2's binding to microtubules". For those reviewers who give their assent, we will be publishing their names alongside the published article.

Nature Structural & Molecular Biology offers a Transparent Peer Review option for new original research manuscripts submitted after December 1st, 2019. As part of this initiative, we encourage our authors to support increased transparency into the peer review process by agreeing to have the reviewer comments, author rebuttal letters, and editorial decision letters published as a Supplementary item. When you submit your final files please clearly state in your cover letter whether or not you would like to participate in this initiative. Please note that failure to state your preference will result in delays in accepting your manuscript for publication.

Cover suggestions

As you prepare your final files we encourage you to consider whether you have any images or illustrations that may be appropriate for use on the cover of Nature Structural & Molecular Biology.

If your image is selected, we may also use it on the journal website as a banner image, and may need

to make artistic alterations to fit our journal style.

Nature Structural & Molecular Biology has now transitioned to a unified Rights Collection system which will allow our Author Services team to quickly and easily collect the rights and permissions required to publish your work. Approximately 10 days after your paper is formally accepted, you will receive an email in providing you with a link to complete the grant of rights. If your paper is eligible for Open Access, our Author Services team will also be in touch regarding any additional information that may be required to arrange payment for your article.

Please note that *Nature Structural & Molecular Biology* is a Transformative Journal (TJ). Authors may publish their research with us through the traditional subscription access route or make their paper immediately open access through payment of an article-processing charge (APC). Authors will not be required to make a final decision about access to their article until it has been accepted. [Find out more about Transformative Journals](https://www.springernature.com/gp/open-research/transformative-journals)

Authors may need to take specific actions to achieve [compliance with funder and institutional open access mandates](https://www.springernature.com/gp/open-research/funding/policy-compliance-faqs). If your research is supported by a funder that requires immediate open access (e.g. according to [Plan S principles](https://www.springernature.com/gp/open-research/plan-s-compliance)) then you should select the gold OA route, and we will direct you to the compliant route where possible. For authors selecting the subscription publication route, the journal's standard licensing terms will need to be accepted, including [self-archiving policies](https://www.springernature.com/gp/open-research/policies/journal-policies). Those licensing terms will supersede any other terms that the author or any third party may assert apply to any version of the manuscript.

Please use the following link for uploading these materials:
[Redacted]

Best regards,

Sophia Frank
Editorial Assistant
Nature Structural & Molecular Biology

nsmb@us.nature.com

On behalf of

Florian Ullrich, Ph.D.
Associate Editor
Nature Structural & Molecular Biology
ORCID 0000-0002-1153-2040

Final Decision Letter:

Message 10th Oct 2022

:
Dear Dr. Reck-Peterson,

We are now happy to accept your revised paper "Structural basis for Parkinson's Disease-linked LRRK2's binding to microtubules" for publication as a Article in Nature Structural & Molecular Biology.

To assist our authors in disseminating their research to the broader community, our SharedIt initiative provides all co-authors with the ability to generate a unique shareable

link that will allow anyone (with or without a subscription) to read the published article. Recipients of the link with a subscription will also be able to download and print the PDF.

Your paper will be published online soon after we receive proof corrections and will appear in print in the next available issue. You can find out your date of online publication by contacting the production team shortly after sending your proof corrections. Content is published online weekly on Mondays and Thursdays, and the embargo is set at 16:00 London time (GMT)/11:00 am US Eastern time (EST) on the day of publication. Now is the time to inform your Public Relations or Press Office about your paper, as they might be interested in promoting its publication. This will allow them time to prepare an accurate and satisfactory press release. Include your manuscript tracking number (NSMB-A45874B) and our journal name, which they will need when they contact our press office.

About one week before your paper is published online, we shall be distributing a press release to news organizations worldwide, which may very well include details of your work. We are happy for your institution or funding agency to prepare its own press release, but it must mention the embargo date and Nature Structural & Molecular Biology. If you or your Press Office have any enquiries in the meantime, please contact press@nature.com.

Please note that *Nature Structural & Molecular Biology* is a Transformative Journal (TJ). Authors may publish their research with us through the traditional subscription access

route or make their paper immediately open access through payment of an article-processing charge (APC). Authors will not be required to make a final decision about access to their article until it has been accepted. [Find out more about Transformative Journals](https://www.springernature.com/gp/open-research/transformative-journals)

Best wishes,

Carolina Perdigoto, PhD
Chief Editor
Nature Structural & Molecular Biology
orcid.org/0000-0002-5783-7106

Click here if you would like to recommend Nature Structural & Molecular Biology to your librarian:

<http://www.nature.com/subscriptions/recommend.html#forms>